# PTCG: Persona-guided Tree-based Counterargument Generation

## Abstract

The ability to generate counterarguments is important for fostering critical thinking, balanced discourse, and informed decision-making. However, existing approaches typically produce only a single counterargument, thereby overlooking the diversity and persuasiveness required in real-world debates. This limitation is critical, as the same topic may persuade different individuals only when framed from distinct perspectives. To address this limitation, we propose Persona-guided Tree-based Counterargument Generation (PTCG), a framework that combines Tree-of-Thoughts–inspired step-wise generation and pruning with speaker persona selection. By estimating the author's persona from the original argument and incorporating speaker personas representing distinct perspectives, the framework operationalizes perspective-taking, enabling reasoning from multiple standpoints and supporting the generation of diverse counterarguments. We propose a tree-based procedure that generates plans, selects the best, and produces multiple speaker persona-specific counterarguments, from which the most effective are chosen. We evaluate PTCG through a comprehensive multi-faceted setup, combining LLM(Large Language Model)-as-a-Judge, classifier-based assessment, and human evaluations. Our experimental results show that PTCG substantially improves both the diversity and persuasiveness of counterarguments compared to baselines. These findings highlight the effectiveness of adaptive persona integration in boosting diversity and strengthening persuasiveness.

## 1 Introduction

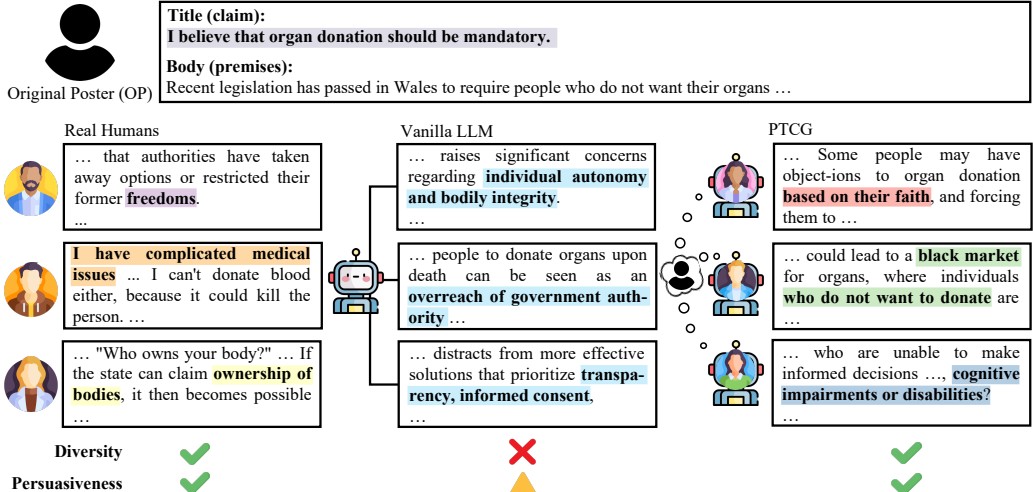

Figure 1: Comparison of LLM-generated counterarguments with and without Persona-guided Tree-based Counterargument Generation (PTCG). While vanilla LLMs tend to produce theoretical and superficial responses that revolve around similar content, PTCG generates counterarguments that reflect diverse perspectives and stronger persuasiveness, more akin to how real people argue. Each color represents a distinct perspective on the topic.

The ability to generate effective counterarguments is a growing area of interest in computational argumentation (Wang et al., 2023). A counterargument is not merely an expression of disagreement but a reasoned response that challenges an argument, exposing assumptions, logical gaps, or alternative perspectives. Engaging with counterarguments is widely recognized as a key mechanism for fostering critical thinking, as it encourages individuals to evaluate evidence, weigh competing viewpoints, and refine reasoning (Ennis, 2015; Dekker, 2020). The ability to generate effective counterarguments is especially important in contexts such as political debates, legal reasoning, and online discussions, where opposing views promote balanced discourse and better-informed decision-making (Li et al., 2020a; Behrendt et al., 2025; Zhang & Ashley, 2025; Gray et al., 2025).

Existing counterargument generation approaches suffer from two critical limitations: First, most methods generate a counterargument focusing on a single point, failing to capture diverse perspectives on the subject. This limitation stems from their reliance on a single strategy, such as attacking a weak premise (Alshomary et al., 2021), directly refuting the conclusion (Alshomary & Wachsmuth, 2023), or pointing out logical flaws (Lin et al., 2023). Second, counterarguments produced by existing approaches, including those generated with large language models (LLMs), often lack persuasiveness. While Chen et al. (2024) demonstrate the potential of LLMs in argument generation, subsequent studies (e.g. Lu et al. 2025, Plenz et al. 2025) reveal that generated arguments often lack value-based reasoning and scenario-driven perspectives essential for persuasiveness in human argumentation. In addition, the aforementioned issue of narrow perspective also negatively impacts the overall persuasiveness of the counterargument. This suggests that persuasiveness across diverse audiences depends on presenting arguments from multiple perspectives, making the resolution of this limitation a central challenge for counterargument generation.

To address these issues, we propose Persona-guided Tree-based Counterargument Generation (PTCG), a framework grounded in the theory of perspective-taking. Perspective-taking, widely studied in social psychology, refers to the practice of imaginatively adopting others' standpoints when evaluating or constructing arguments. It has been shown to foster empathy, reduce bias, and encourage reasoning from viewpoints different from one's own (Batson et al., 1997; Green & Brock, 2000). PTCG operationalize perspective-taking by guiding counterargument generation with pre-defined personas. Specifically, after estimating the original author's persona from their argument, the framework selects personas from both similar and contrasting predefined persona clusters and uses them to guide counterargument generation. The predefined persona clusters were created to organize about 50,000 personas, reducing redundancy and enabling efficient selection of diverse perspectives. This design enables models to move beyond default stances and generate counterarguments that reflect a wider range of perspectives. In addition, PTCG incorporates a Tree-of-Thoughts (ToT)–inspired reasoning procedure (Yao et al., 2023). Multiple candidate reasoning paths called *plans* for generating counterarguments are first generated, evaluated, and pruned. Only the most promising plans are then expanded into full counterarguments. This iterative process of generation and selection allows for diverse and persuasive reasoning paths. As illustrated in Figure 1, employing PTCG can improve the diversity and persuasiveness of generated counterarguments. By combining persona conditioning with a step-wise reasoning process inspired by Tree-of-Thoughts (ToT) (Yao et al., 2023), the framework iteratively generates and selects candidate plans and counterarguments, ultimately producing multiple counterarguments that capture both diversity and persuasiveness.

To evaluate PTCG, we conduct experiments with multiple LLMs using 847 discussion threads from the ChangeMyView subreddit, which cover a diverse range of real-world topics. We combine LLM-as-a-Judge to assess the diversity and persuasiveness—general and targeted—as well as the stance and quality of the generated counterarguments. We further incorporate classifier-based metrics for the key dimension of persuasiveness, providing a more comprehensive evaluation. In addition, we conduct human evaluation, which not only complements the LLM-as-a-Judge results but also demonstrates persuasiveness across a diverse pool of evaluators, providing further evidence of applicability to the real audience. Across these evaluations, PTCG consistently outperforms the baselines, producing counterarguments that are more diverse, persuasive, and higher in overall quality.

## 2 RELATED WORK

**Argument Generation** Early work on argument generation framed the task as a largely symbolic or rule-driven process. Sato et al. (2015) proposed a debating system that generates arguments through

a pipeline of rule-based modules, such as topic analysis, evidence retrieval, and template-based surface realization. While this line of work demonstrates that fully automatic argument generation is feasible, the resulting systems tend to be brittle and difficult to scale beyond predefined domains and templates. Wachsmuth et al. (2018) moved toward more flexible generation while still relying on explicitly modeled rhetorical strategies. Their system composes arguments using hand-crafted rhetorical patterns, showing that explicit control over argumentative structure can improve coherence and persuasiveness. Hua & Wang (2018) introduced neural argument generation augmented with retrieved evidence, combining neural text generation with external document retrieval to ground arguments in factual content. Together, these approaches laid important groundwork for argument and counterargument generation. However, they provide limited control over who is speaking (persona) and offer little support for generating diverse perspectives across outputs.

**Counterargument Generation** Recent studies on counterargument generation have mainly focused on explicit argument structures or strategies. For example, Alshomary et al. (2021) propose attacking weak premises, while Alshomary & Wachsmuth (2023) guide generation by simultaneously modeling the conclusion of the original post. Lin et al. (2023) instead operate at the sentence level, producing concise counterarguments for each statement. However, these approaches rely on a single strategy, offering limited opportunities to reason from the opponent's perspective and typically producing only one counterargument. In contrast, we propose a method that overcomes these limitations by generating multiple counterarguments that reflect diverse perspectives of the opponent. Hu et al. (2025) proposes a persona-driven multi-agent framework in which multiple artificial personas engage in a debate and their discussion is merged into a single argumentative essay. In contrast, our task formulation is fundamentally different: instead of synthesizing multiple voices into one unified output, our framework generates multiple independent and persona-conditioned counterarguments, with controlled diversity across personas being a core objective rather than a byproduct of debate.

**Perspective-Taking** Psychological studies highlight the persuasive power of perspective-taking and narrative immersion. Batson et al. (1997) show that imagining how others feel fosters empathy and altruistic motivation. Green & Brock (2000) and Mar & Oatley (2008) suggest that narrative "transportation" enables simulated experience, which can influence attitudes more deeply than factual exposition. More recently, Bullock et al. (2021) argue that narratives are persuasive partly because they are processed more fluently than non-narrative formats. These findings motivate our use of perspective-taking-based generation to induce perspectival engagement and simulate meaningful disagreement. Moreover, perspective-taking enables the incorporation of diverse viewpoints, making it possible to generate multiple counterarguments for a single post.

**Diverse-Audience Persuasion** Lukin et al. (2017) show that persuasiveness depends on audience traits, motivating the use of personality-based analysis in argumentation. Building on this, studies have explored personalization in persuasive dialogue: Wang et al. (2019) adapt strategies based on user traits, and Al Khatib et al. (2020) incorporate debaters' characteristics to improve persuasiveness prediction. Recent work further demonstrates that LLMs can generate more persuasive messages when tailored to psychological profiles (Matz et al., 2024), modulate linguistic features according to personality cues (Mieleszczenko-Kowszewicz et al., 2024), and role-play personas to enhance empathy and strategy distribution (Yang et al., 2025b). Building on this line of research, our work develops an approach that generates diverse counterarguments with awareness of audience diversity, aiming for greater persuasiveness.

# 3 PERSONA-GUIDED TREE-BASED COUNTERARGUMENT GENERATION

## 3.1 TASK DESCRIPTION

We define the task of **multiple distinct counterargument generation** to evaluate the ability to generate persuasive counterarguments that cover a broad spectrum of perspectives on a given argument. Specifically, the input is an *argument* consisting of a claim and one or more premises supporting it, and the output is a set of *counterarguments*, each presenting a distinct perspective that challenges the original argument. We detail the evaluation criteria in Section 4.3

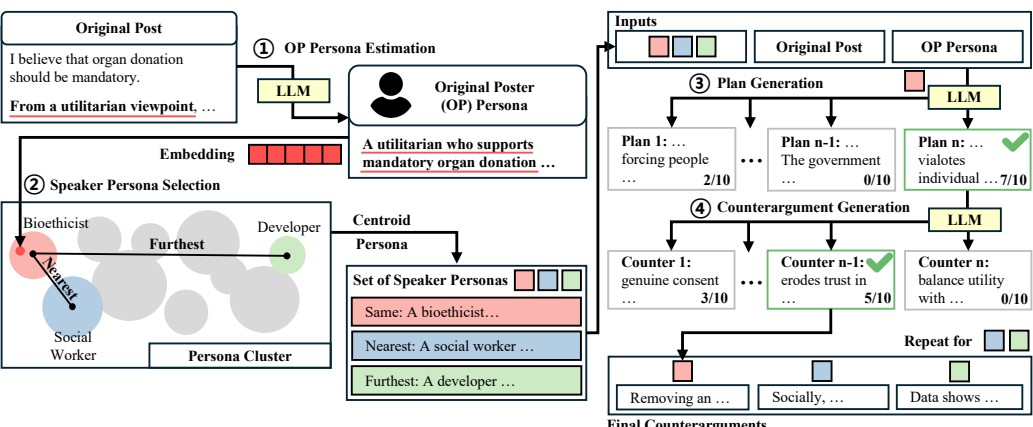

Figure 2: Persona-guided Tree-based Counterargument Generation (PTCG). An LLM first extracts the author's persona and conditions on distinct personas to produce multiple candidate reasoning plans. These plans are evaluated and pruned in a tree-based step-wise manner, and the most promising ones are expanded into three final counterarguments, each grounded in a different persona.

## 3.2 PERSPECTIVE TAKING: PERSONA SELECTION

To realize perspective-taking in counterargument generation, we draw on the notion of personas as proxies for diverse viewpoints. Prior work has shown that incorporating personas into dialogue models allows them to go beyond generic language modeling, producing responses that are more consistent, human-like, and reflective of underlying experiences or viewpoints (Zhang et al., 2018; Moon et al., 2024). We adopt the PersonaHub dataset (Ge et al., 2024), a large-scale repository of about 50,000 personas. Since using all personas is redundant and costly, we cluster similar ones into groups. Each persona is embedded using OpenAI's text-embedding-3-large (OpenAI 2024; 3,072-dimensional), reduced to a 50-dimensional representation via UMAP (McInnes et al., 2018), and clustered into 39 groups[1] using HDBSCAN (McInnes et al., 2017). Subsequently, we leverage inter-cluster distances to select counterargument speakers from distinct groups, ensuring that the generated counterarguments embody genuinely different perspectives. To help readers better understand the semantic coherence and diversity encoded in these clusters, we provide representative persona examples from several clusters in Section A2. These examples illustrate how grouping similar personas enables systematic selection of viewpoints that truly differ in background, expertise, and worldview—an essential property for perspective-taking in counterargument generation.

## 3.3 PERSONA-GUIDED TREE-BASED COUNTERARGUMENT GENERATION (PTCG) FRAMEWORK

Our method, which we call Persona-guided Tree-based Counterargument Generation (PTCG; See Figure 2 and Algorithm 1), integrates the Original Poster (OP) persona estimation, clustering-based speaker persona selection, and a Tree-of-Thoughts (ToT) (Yao et al., 2023)-inspired step-wise generation process (Plan and Counterargument Generation and Selection). For all stages, we employ *Llama-3.1-8B-Instruct* as the underlying language model.

**Step 1: Original Poster (OP) Persona Estimation**  Given an original post, we estimate the OP persona using an LLM-based estimation prompt. Here, the OP refers to the author of the original post. This estimated persona represents the OP's beliefs, values, and worldview, and conditions subsequent counterargument generation. The detailed prompt is provided in Appendix Figure A3.

**Step 2: Speaker Persona Selection**  To balance diversity with interpretability, we set the number of personas to three, supported by observations from the CMV dataset where posts with multi-

---

[1]As shown in Appendix Table A8, the configuration with 39 clusters was selected as it achieves the highest Silhouette Score (Rousseeuw, 1987) while also maintaining a strong Calinski–Harabasz Index (Caliński & Harabasz, 1974), indicating the best balance between cohesion and separation.

---

**Algorithm 1** Persona-guided Tree-based Counterargument Generation (PTCG)

---

**Require:** Original post $x$, persona cluster centroids $\mathcal{C}$
**Ensure:** Final counterarguments $\mathcal{Y} = \{y_1, y_2, y_3\}$

1: **Step 1: Original Poster (OP) Persona Estimation**
2: $p_{\mathrm{op}} \leftarrow \texttt{LLM\_Estimate\_Persona}(x)$          ▷ Prompt: Figure A3
3: $c_{\mathrm{op}} \leftarrow \texttt{Nearest\_Centroid}(p_{\mathrm{op}}, \mathcal{C})$
4:
5: **Step 2: Speaker Persona Selection**
6: Select three speaker personas from $\mathcal{C}$:
     $p_{\mathrm{same}} = $ centroid persona from $c_{\mathrm{op}}$          ▷ Same cluster
     $p_{\mathrm{nearest}} = $ centroid persona from nearest cluster
     $p_{\mathrm{furthest}} = $ centroid persona from furthest cluster
7: $\mathcal{P}^{\star} = \{p_{\mathrm{same}}, p_{\mathrm{nearest}}, p_{\mathrm{furthest}}\}$
8:
9: **Step 3: Plan Generation and Selection**
10: **for** each $p \in \mathcal{P}^{\star}$ **do**
11:     $\{r_1, r_2, r_3\} \leftarrow \texttt{LLM\_Generate\_Plans}(x, p_{\mathrm{op}}, p)$      ▷ Prompt: Figure A4
12:     $r_p^{\star} \leftarrow \texttt{LLM\_Select\_Plan}(\{r_1, r_2, r_3\}, p, p_{\mathrm{op}})$      ▷ Prompt: Figure A5
13: **end for**
14:
15: **Step 4: Counterargument Generation and Selection**
16: **for** each $p \in \mathcal{P}^{\star}$ **do**
17:     $\{y_1^p, y_2^p, y_3^p\} \leftarrow \texttt{LLM\_Generate\_Counters}(x, p_{\mathrm{op}}, p, r_p^{\star})$      ▷ Prompt: Figure A4
18:     $y_p^{\star} \leftarrow \texttt{LLM\_Select\_Counter}(\{y_1^p, y_2^p, y_3^p\}, p, p_{\mathrm{op}})$      ▷ Prompt: Figure A5
19:     Add $y_p^{\star}$ to $\mathcal{Y}$
20: **end for**
21:
22: **return** $\mathcal{Y}$

---

ple delta-awarded comments—indicating diverse, high-quality counterarguments—rarely exceeded three (see Appendix Figure A2). We select personas based on their cluster distance from the OP: one from the *same* cluster, one from the *nearest* cluster, and one from the *furthest* cluster. For each cluster, no persona exactly matched the centroid; therefore, we used the closest in embedding space. This setup ensures that the three chosen personas collectively reflect perspectives ranging from highly aligned to markedly divergent, thereby systematically probing how cluster distance influences counterargument generation and persuasiveness.

**Step 3: Plan Generation and Selection**   Once the OP persona and speaker personas are determined, the generation process proceeds in a tree-based step-wise manner inspired by Tree-of-Thoughts (Yao et al., 2023). For each persona, the LLM generates three candidate plans, each outlining a persuasive strategy for counterargument generation. A voting procedure then evaluates these candidates in terms of whether they effectively use the contrast between personas, apply a strong strategy, and present their reasoning clearly and logically. The most promising plan is selected among the three, resulting in one best plan per persona. For each speaker persona, one finalized plan is thus determined and passed to the subsequent generation stage. The prompts used for plan generation and selection can be found in Figure A4 and Figure A5, respectively.

**Step 4: Counterargument Generation and Selection**   Building on the selected plans, the finalized plan for each speaker persona is explicitly used, along with the original post, OP persona, and the designated speaker persona, to guide counterargument generation. The LLM then generates three candidate counterarguments per persona, each following the selected plan while reflecting the persona's distinct perspective. A voting procedure evaluates these candidates based on whether they leverage the contrast between personas, directly challenge the original argument, and are specific, persuasive, and logically consistent. Among the three candidates, the most persuasive counterargument is selected as the final output for that persona. This process is repeated for each persona, with one best counterargument selected per persona, yielding three diverse counterarguments from the

same, nearest, and furthest clusters that collectively capture multiple perspectives. The prompts for this step are identical to those in Step 3, shown in Figure A4 and Figure A5, respectively.

# 4 EXPERIMENTS

## 4.1 DATASET

To assess the ability to generate diverse and persuasive counterarguments, we constructed a dataset of 847 *ChangeMyView* (CMV) subreddit[2] posts (i.e., arguments), each paired with three comments (i.e., counterarguments) that have successfully persuaded the original poster as gold-standard persuasive counterarguments. On CMV, each post consists of a title summarizing the main claim and a body providing the premises supporting the claim. Original posters award a delta ($\Delta$) to comments that successfully change their view, which we treat as quality counterarguments. We first collected 72,999 posts from CMV, spanning the years 2013 to 2023. Then, to ensure diversity, we filter for posts with three delta-awarded comments, yielding a dataset of 847 post-comments pairs.

## 4.2 BASELINES

To establish meaningful points of comparison, we adopt baselines from both prior counterargument generation research and representative LLMs.

**Argument Undermining (Alshomary et al., 2021)** This method identifies weak premises in the original post and generates counterarguments by attacking them. For fair comparison, we first employ a weak-premise identification model to select the top three weak premises, then generate one counterargument per premise, resulting in three outputs.

**Joint One-seq (Alshomary & Wachsmuth, 2023)** This method infers multiple conclusions from the premises in the original post and uses them as the basis for counterargument generation. We sample three alternative conclusions and generate a counterargument for each.

**DeepSeek-R1 (Guo et al., 2025)** A reasoning-optimized model trained with reinforcement learning to improve logical consistency. It serves as a comparison to test whether explicit reasoning alone, without persona guidance, improves argumentative coherence. For counterargument generation, we prompt the model to produce three outputs in one pass, as shown in Figure A6.

**Llama 3.1 (Grattafiori et al., 2024)** An instruction-tuned model from the Llama 3.1 family trained on large-scale publicly available datasets. It is widely employed as a general-purpose baseline for zero-shot inference due to its balance between model size and reasoning capability. We use the *Llama-3.1-8B-Instruct* model, which also serves as the backbone for our proposed method. For comparability, we follow the same prompting strategy as DeepSeek-R1.

## 4.3 EVALUATION METRICS

We evaluate our method across four complementary dimensions: Persuasiveness, Perspective Diversity, and Stance. We employ *LLM-as-a-Judge* (Zheng et al., 2023; Gu et al., 2024), as it provides more reliable evaluations than traditional lexical-overlap-based metrics such as BLEU and ROUGE, which often diverge from human judgment (Celikyilmaz et al., 2020; Hu et al., 2024). For targeted persuasiveness, which measures how well the counterarguments resonate with the original poster, we additionally use data-driven *classifier-based scores*.

**Persuasiveness** While general persuasiveness is important, it is equally crucial for counterarguments to be persuasive with respect to the original poster's context. To capture both, we evaluate *general persuasiveness* using only the title (claim) of the original post, and *targeted persuasiveness* using both the title and body (premises), which assesses resonance with the original poster's context and simulates realistic dialogue. Since each input yields multiple counterarguments, we report the average persuasiveness score to reflect overall persuasiveness. The prompts used for these two evaluations are provided in Appendix Figures A7 and A8.

---

[2] https://www.reddit.com/r/changemyview/

Table 1: Evaluation results of baselines and PTCG measured by GPT-4o-mini. Reported scores have been averaged over five runs. PTCG (Ours) is implemented on top of the Llama-3.1-8B-Instruct model. *Persuasiveness* is reported for general and targeted settings, scored on a 1–10 scale. *Perspective diversity* evaluates the variety of viewpoints reflected in generated counterarguments, using a 1–5 scale. For *quality*, we report appropriateness (App.), clarity (Cla.), grammaticality (Gra.), and relevance (Rel.), each on a 1–5 scale. *Stance* measures alignment with the opposite stance, reported on a 0–100 scale. Best scores are shown in bold.

| Approach | Persuasiveness | | Perspective Diversity | Quality | | | | Stance |
|---|---|---|---|---|---|---|---|---|
| | General | Targeted | | App. | Cla. | Gra. | Rel. | |
| Argument Undermining | 3.48 | 2.85 | 1.84 | 1.72 | 1.55 | 1.60 | 1.59 | 82.12 |
| Joint One-seq | 3.51 | 2.85 | 1.86 | 1.74 | 1.60 | 1.68 | 1.55 | 78.01 |
| DeepSeek-R1 | 8.05 | 7.26 | 4.13 | 4.43 | 4.23 | 4.92 | 4.65 | 83.14 |
| Llama 3.1 | 8.07 | 7.20 | 4.21 | 4.44 | 4.34 | **4.98** | 4.65 | 84.04 |
| PTCG (Ours) | **8.26** | **7.42** | **4.27** | **4.54** | **4.44** | **4.98** | **4.76** | **85.10** |

**Perspective Diversity** Producing counterarguments that embody genuinely distinct viewpoints—beyond superficial rewordings—is difficult. True diversity requires capturing ideological, emotional, or experiential variation that reflects distinct perspectives. We therefore evaluate whether the outputs move beyond lexical variation and invoke deeper interpretive frames not explicitly given in the input. The prompt for evaluating perspective diversity is provided in Appendix Figure A9.

**Quality** Generating counterarguments that are grammatically correct, logically coherent, and contextually appropriate remains a fundamental challenge, as large language models often produce verbose or vague content that undermines clarity and argumentative strength. To comprehensively assess this dimension, we adopt four established criteria grounded in prior research on argumentation quality: *grammaticality*, *appropriateness*, *relevance*, and *clarity*. The rationale and further details for selecting these dimensions are provided in Appendix Section A3. The prompt used for evaluating quality is provided in Appendix Figure A10.

**Stance** Ensuring that all generated counterarguments clearly oppose the original post is essential. However, LLMs may sometimes produce neutral or contradictory outputs unless properly guided, particularly when generating multiple responses. We therefore measure how reliably each method produces outputs that both contradict the original post and remain contextually appropriate. The prompt used for evaluating stance is provided in Appendix Figure A11.

## 5 RESULTS AND ANALYSIS

### 5.1 RATING-SCALE EVALUATION

The reported scores were determined by averaging across five runs to control for random variation. Table 1 shows that PTCG consistently outperforms all baselines across all evaluation dimensions. The general and targeted persuasiveness scores demonstrate that the counterarguments generated by PTCG are effective for a general audience, as well as the particular original poster. The outcome for perspective diversity confirms the ability to produce counterarguments that reflect diverse viewpoints. The performance on the quality metrics shows improved overall textual quality. Lastly, the results on stance indicate that the generated counterarguments more clearly oppose the respective original posts. Overall, PTCG consistently outperforms the baselines across persuasiveness, perspective diversity, textual quality, and stance clarity.

### 5.2 CLASSIFIER-BASED EVALUATION

To complement LLM-based judgments, we use a Delta classifier trained on CMV delta-award annotations to assess targeted persuasiveness. Results are shown in Table A9. Table 2 shows that PTCG achieves the highest delta score of 0.82, clearly outperforming all baseline models. This superiority extends to both strong LLMs and earlier counterargument generation methods, underscoring the consistent advantage of our approach.

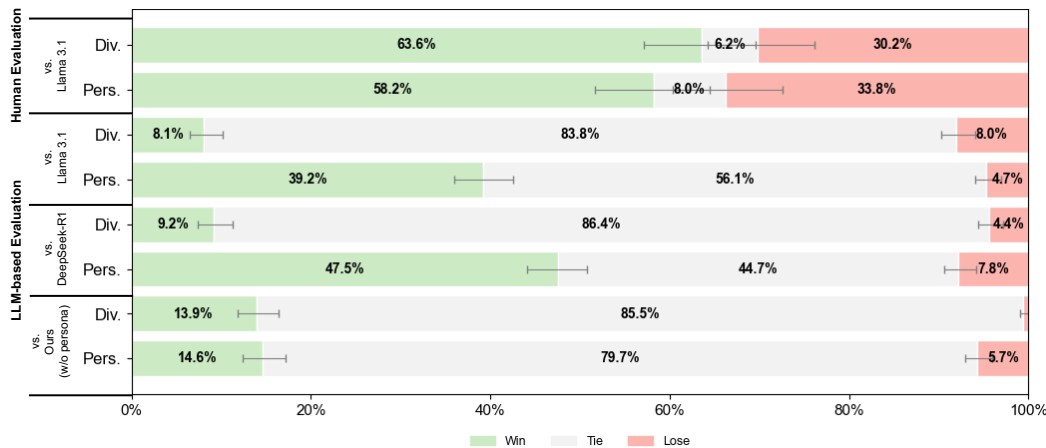

Figure 3: Win/Tie/Lose analysis of diversity (Div.) and persuasiveness (Pers.) across Human Evaluation (top) and LLM-based Evaluation using GPT-4o-mini (bottom). The figure compares our method (PTCG) with baselines, showing the proportion of cases where each system's counterarguments were judged as more diverse or persuasive (Win), equivalent (Tie), or less effective (Lose).

These results confirm that our approach generates counterarguments that are consistently more persuasive. Notably, this evaluation is conducted through a classifier trained on real-world data. The results are well aligned with those from the LLM-based targeted persuasiveness evaluation, further validating the robustness of our findings.

### 5.3 PAIRWISE EVALUATION

To assess our framework's effectiveness, we conduct human and LLM-based pairwise evaluations, focusing on the persuasiveness and diversity of generated counterarguments.

Table 2: Targeted persuasiveness (Delta score) is evaluated using the delta classifier. The score ranges from –1 to 1 and represents the predicted probability of receiving a delta. For clarity, the best results are highlighted in bold.

| Approach | Delta Score |
|---|---|
| Argument Undermining | -0.64 |
| Joint One-seq | -0.12 |
| DeepSeek-R1 | 0.80 |
| Llama 3.1 | 0.78 |
| PTCG (Ours) | **0.82** |

For the human evaluation, we randomly sampled 95 inputs and asked five human evaluators to rate the outputs from PTCG and the baselines. Details are in Appendix Section A4. As shown in As shown in As shown in Figure 3, human judges consistently rated PTCG higher than the baseline across both evaluation dimensions.[3] Since both PTCG and the baseline rely on the same Llama 3.1 model, this constitutes a fair comparison and demonstrates that the performance gains come directly from our framework rather than differences in model capacity or training data. In particular, PTCG generated counterarguments that reflected a wider variety of perspectives, which contributed to greater diversity. This diversity, in turn, suggests an enhanced capacity to appeal to and persuade a broader range of audiences. Moreover, the persuasiveness gains observed in PTCG indicate the evaluators found the generated counterarguments not only more varied but also more compelling. Taken together, these results reinforce that diversity and persuasiveness are complementary rather than competing qualities: by incorporating multiple perspectives, PTCG achieves persuasive power that resonates with different evaluators.

We also conducted pairwise comparisons using GPT-4o-mini as the evaluator, assessing persuasiveness and diversity. To mitigate potential ordering effects (Zheng et al., 2023), we presented the counterargument sets in both orders: if the evaluation was first conducted in the AB order, it was also repeated in the BA order, and the final scores were aggregated accordingly. We also allowed for *Tie*, which capture cases where the evaluator either explicitly selected a "Hard" option or produced

---

[3]Inter-annotator agreement, assessed via Fleiss' Kappa (0.619) and Krippendorff's Alpha (0.612), reflects substantial consistency among annotators.

inconsistent preferences when the order of the two sets was reversed, reflecting potential ordering effects inherent in LLM-based pairwise judgments. Figure 3 shows the results comparing PTCG against both LLM baselines, Llama 3.1 and DeepSeek-R1. Across all comparisons, PTCG shows consistent improvements in both diversity and persuasiveness. For diversity, while baseline models show very high tie rates, our method secures substantially higher win rates, indicating that it is able to broaden the argumentative space beyond what the baselines typically generate. Compared to our ablation without persona (w/o persona) conditioning, PTCG achieves nearly double the win rate, underscoring the critical role of personas in enhancing diversity by injecting distinct perspectives. For persuasiveness, the improvements are even more pronounced. Against Llama 3.1, PTCG achieves nearly an eightfold higher win rate and against DeepSeek-R1 it achieves a sixfold higher win rate.

These results suggest that grounding counterarguments in diverse personas significantly enhances their ability to engage and convince. Compared to the ablation w/o persona conditioning, the win rate improvement is smaller but remains consistent. This consistency highlights that grounding counterarguments in diverse personas still plays a meaningful role in enhancing persuasiveness. Taken together, these findings reinforce that PTCG does not simply generate more varied content but also produces counterarguments that evaluators consistently judge as more compelling. The high tie rates across baselines reflect the difficulty of the task and the nuanced nature of pairwise judgments, but the consistent win margins achieved by our method indicate robust and reliable improvements.

## 5.4 QUALITATIVE ANALYSIS

Analysis of the generated counterarguments[4] reveals that, unlike the LLM baselines, PTCG incorporates rich social, cultural, and practical contexts while emphasizing concrete risks and lived human experiences. This makes the outputs more persuasive and relatable. Baseline outputs often reiterate high-level ethical tropes such as autonomy, government overreach, or informed consent without moving beyond abstract formulations. In contrast, PTCG highlights tangible concerns such as potential inequalities in medical access, the emergence of black markets, and the coercive treatment of bodies against deeply held cultural or spiritual beliefs. A similar pattern is observed in the movie theater example, where PTCG draws attention to broader social and economic factors—such as the role of public spaces in family routines, disproportionate burdens on low-income households, and community-level impacts on local businesses—that baseline models overlook. These dimensions anchor the counterarguments in scenarios that audiences can more readily imagine and evaluate, thereby enhancing their real-world salience. Furthermore, PTCG broadens the argumentative space by drawing from diverse personas that introduce distinct vantage points, ranging from societal fairness to cultural integrity and professional identity. This multiplicity of perspectives not only increases diversity but also strengthens persuasiveness by appealing to different values and lived experiences. In some cases, narrative elements — such as imagining marginalized individuals or communities facing systemic disadvantages — add an additional layer of resonance absent from baseline generations. Consequently, PTCG achieved stronger performance in both diversity and persuasiveness. This improvement stems from not only broadening the range of perspectives but also grounding its arguments in concrete, human-centered reasoning that baseline models fail to capture. In addition to these qualitative differences, our framework naturally produces longer outputs due to its multi-stage, persona-grounded reasoning process.[5] To illustrate why this occurs, we provide qualitative examples showing how different personas lead to distinct argumentative framings, levels of elaboration, and domain-specific reasoning styles. These excerpts demonstrate how persona conditioning influences the depth, structure, and emphasis of each counterargument—ultimately contributing to richer and more detailed responses compared to baseline models.

## 5.5 ABLATION STUDY

We further analyze the contribution of each component through ablation experiments, which evaluate each module in isolation and examine how their combination yields complementary performance gains. Beginning with the baseline Llama 3.1 model, we incrementally add the persona-grounding module, the tree-based step-wise generation module, and a CoT-based step-wise variant (Table 3).

---

[4]Please refer to Table A6 and Table A7 examples.

[5]See Table A5 for detailed length statistics.

Table 3: Ablation study of PTCG. *Persuasiveness* is reported for general and targeted settings, scored on a 1–10 scale. *Perspective diversity* evaluates the variety of viewpoints reflected in generated counterarguments, using a 1–5 scale. For *quality*, we report appropriateness (App.), clarity (Cla.), grammaticality (Gra.), and relevance (Rel.), each on a 1–5 scale. *Stance* measures alignment with the opposite stance, reported on a 0–100 scale. For clarity, best results are in bold.

| Configuration | Persuasiveness | | Perspective Diversity | Quality | | | | Stance |
|---|---|---|---|---|---|---|---|---|
| | General | Targeted | | App. | Cla. | Gra. | Rel. | |
| Llama 3.1 | 8.07 | 7.20 | 4.21 | 4.44 | 4.34 | 4.98 | 4.65 | 84.04 |
| + Persona | 7.76 | 6.72 | 4.20 | 4.22 | 4.20 | 4.92 | 4.32 | 83.80 |
| + CoT-based Gen. | 8.03 | 7.39 | 4.26 | 4.52 | 4.43 | 4.98 | 4.74 | 84.98 |
| + Tree-based Gen. | 8.24 | **7.50** | 4.03 | **4.65** | **4.56** | **4.99** | **4.88** | **85.51** |
| + Persona + Tree-based Gen. (PTCG) | **8.26** | 7.42 | **4.27** | 4.54 | 4.44 | 4.98 | 4.76 | 85.10 |

The tree-based generation module has the strongest impact, substantially improving quality metrics and achieving the highest targeted persuasiveness. Its structured exploration of multiple reasoning paths enhances coherence and opponent relevance. However, exploring only the most promising branches reduces perspective diversity, falling below the baseline.

The CoT-based variant shows a different trade-off: it moderately improves persuasiveness and quality while maintaining higher diversity than the tree-based module. This suggests that linear reasoning encourages elaboration without overly constraining the argumentative trajectory, though it remains clearly weaker than tree-based generation in targeted persuasiveness and stance alignment.

Persona grounding alone decreases persuasiveness and offers limited quality gains, but it preserves diversity—indicating that persona signals introduce variation but require structured reasoning to be effective. When paired with tree-based reasoning, persona grounding offsets the diversity loss, leading to the highest diversity while maintaining competitive persuasiveness.

Overall, the full configuration (PTCG) yields the most balanced outcome: the strongest general persuasiveness, competitive targeted persuasiveness, the highest diversity, and consistently strong quality and stance alignment. These results show that tree-based reasoning provides structural rigor, CoT offers lightweight gains, and persona grounding introduces diverse viewpoints—together enabling PTCG to produce counterarguments that are both highly persuasive and meaningfully diverse.

# 6    CONCLUSION AND FUTURE WORK

In this work, we proposed and addressed the task of generating multiple distinct and persuasive counterarguments grounded in realistic personas. Inspired by Tree-of-Thoughts approach, we adopted a tree-based step-wise generation with pruning process to enhance the quality of generated content, while integrating personas based on distance-based selection. This design enables our approach to overcome the inherent limitations of base LLMs and produce a broader range of compelling counterarguments. For evaluation, we combined LLM-based judgments, and classifier-based assessments, providing a comprehensive multi-faceted validation of our method. The results demonstrate that our persona-grounded, tree-based step-wise generation approach significantly improves both the diversity and persuasiveness of counterarguments. Our work provides a new direction for counterargument generation research. It also suggests practical applicability in fostering critical thinking, facilitating balanced debates, and supporting informed decision-making.

We show that PTCG improves the generation of diverse and persuasive counterarguments, and there are several promising directions for future work. First, PTCG focuses on generating counterarguments for a single opinion. However, debates often unfold over multiple rounds, with each round consisting of an exchange between participants (Durmus & Cardie, 2019; Li et al., 2020b). Extending PTCG to multi-round debates would enable models to engage in interactive and dynamic exchanges, where counterarguments are refined over multiple turns. This line of research could further extend to multi-party debates, where multiple participants interact and compete (Sia et al., 2022). Second, PTCG is limited to text-based personas derived from clustering. Incorporating richer user signals, such as value-based attributes, may further improve the persuasiveness of generated counterarguments (Lukin et al., 2017). We plan to explore these directions in our future work.

ETHICS STATEMENT

This study makes use of the publicly available Reddit ChangeMyView dataset, which has also been widely adopted in prior work on computational argumentation. The dataset was used strictly for research purposes. We are aware that generating persuasive counterarguments with large language models raises ethical concerns, particularly the risk of misuse in manipulative or coercive contexts. To reduce these risks, our work is limited to academic exploration, with the goal of examining diversity of perspectives rather than promoting adversarial persuasion. In addition, to prevent the risk of outputs being mistaken for human speech in deceptive ways, we restricted the use of first-person generation. We also emphasize the need for future work to consider safeguards and responsible use practices when deploying such systems.

REPRODUCIBILITY STATEMENT

For the reviewing process, we have submitted an anonymized supplementary zip file containing the full source code and data necessary to reproduce our experiments. This package ensures that reviewers can replicate the reported results, and the code will be made publicly available upon acceptance.[6]

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

## A APPENDIX

### A1 LLM USAGE

We employed GPT-4o to support the literature review and manuscript preparation process. Specifically, the model was used to assist in identifying relevant prior work and refining the clarity and readability of the draft. This usage was limited to auxiliary scholarly support—such as improving grammar and reducing stylistic inconsistencies.

### A2 REPRESENTATIVE PERSONA EXAMPLES BY CLUSTER

To illustrate how the persona clusters are organized, we present representative examples from three clusters: (1) Sports and Physical Education, (2) Finance and Marketing, and (3) Historians. These examples demonstrate the semantic coherence within each cluster, reflecting domain-specific interests, professional backgrounds, and characteristic reasoning patterns.

**Sports and Physical Education**

- A high school physical education teacher seeking to incorporate Paralympic history and achievements into the curriculum to inspire and educate students about inclusivity in sports.
- A sports scientist researching the biomechanics and physics of tennis, focusing on how racket specifications impact performance and injury risks.
- A sports journalist covering the history of ice hockey and its impact on national identity in Poland.
- An elementary school teacher who enjoys incorporating diverse sports stories in her curriculum to inspire students.
- A football coach seeking to learn from successful strategies and team management in various leagues.

**Finance and Marketing**

- A financial analyst specializing in Asian markets and wealthy individuals, interested in tracking the investments and philanthropic activities of billionaires like Gerald Chan.
- A quantitative analyst with expertise in financial modeling and algorithmic trading, seeking to develop and implement systematic value investment strategies.
- A digital marketing specialist interested in innovative aggregator models that consolidate search results from multiple sources.
- A marketing specialist for a tech company, looking for innovative ways to engage with pop culture and fandoms to promote new products and services.
- A business strategist for Arriva UK Bus, interested in exploring opportunities and challenges related to subsidiary operations and company restructuring.

**Historians**

- An Iowa historian focusing on the development and growth of townships in Jones County.
- A historian specializing in 19th-century British architecture, with a focus on the works of notable architects in Lancashire.
- A local historian specializing in the political and business development of Marlborough, Massachusetts in the 19th century.
- A historian specializing in the late medieval and early modern history of France and the Iberian Peninsula, with a focus on power dynamics, family strategies, and women's roles in politics.
- A local historian or genealogist researching the history of small communities and families in Fremont County, Iowa.

## A3    LLM-BASED EVALUATION: QUALITY

Prior studies have established diverse criteria for evaluating the quality of arguments. Alshomary et al. (2021) consider grammaticality and content richness as key factors in assessing generated arguments, while Lin et al. (2023) emphasize appropriateness, grammaticality, and logic, aligning GPT-based and human evaluation through shared criteria. Moreover, Wachsmuth et al. (2017) provide a comprehensive taxonomy of argument quality, highlighting clarity, appropriateness, and Relevance as core components under effectiveness and reasonableness. Based on these findings, we adopt four criteria for LLM-based evaluation of counterarguments: appropriateness, clarity, grammaticality, and relevance.

**Appropriateness:** Whether the language and tone are suitable for the context and proportional to the issue. Inappropriate tone (e.g., overly aggressive or dismissive) lowers the score.

**Clarity:** Whether the writing is clear, well-organized, and free from ambiguity or unnecessary complexity, allowing the reader to easily follow the reasoning.

**Grammaticality:** Whether the text follows standard grammar conventions, including punctuation, sentence structure, and syntax. This ensures the counterargument reads fluently without errors.

**Relevance:** How directly the counterargument engages with the original post and addresses its key points. Superficial or off-topic content would reduce relevance.

## A4    HUMAN EVALUATION

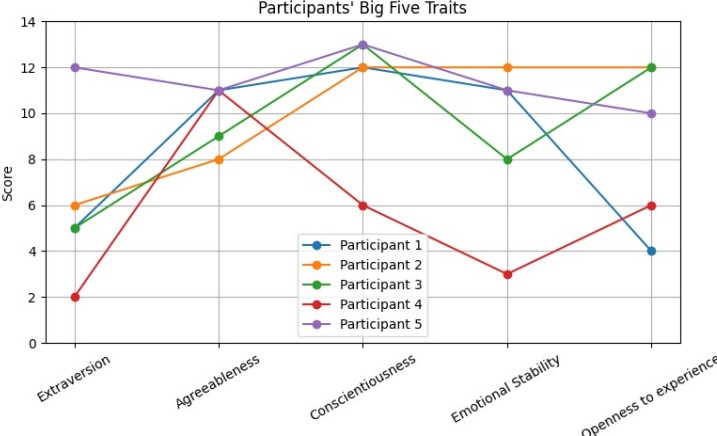

Figure A1: Big Five Personality trait distribution of recruited evaluators. The heterogeneous profiles helped ensure diverse perspectives in human evaluation.

**Recruitment**    We recruited five evaluators comprising both undergraduate and graduate students. In selecting participants, we referred to prior work showing that argument persuasiveness can be influenced by individual personality traits (Lukin et al., 2017), particularly as measured by the Big Five Personality framework. To ensure diversity in perspectives, participants completed the Ten Item Personality Inventory (TIPI) survey (Gosling et al., 2003), and we considered the distribution of their scores during recruitment. This ensured a heterogeneous pool of evaluators, encompassing a wide range of personality traits, as illustrated in Figure A1.

**Procedure**    Each participant reviewed several original posts and corresponding counterarguments generated by different methods. Inspired in part by the evaluation setup of Chung et al. (2025), where annotators assessed outputs on quality and diversity in set-based presentations, we adopted a similar approach. Specifically, from the full test set we sampled 95 instances, and participants were asked to perform two tasks:

In the *persuasiveness task*, participants selected the most persuasive counterargument for each original post. In the *diversity task*, participants compared two sets of counterarguments and judged which set demonstrated greater perspective diversity.

| Model | Avg end-to-end latency/OP (s) | Per response (s) | Percentage (%) |
|---|---|---|---|
| Llama 3.1 (n=1) | $3.16 \pm 0.033$ | $3.16 \pm 0.033$ | 100 |
| Llama 3.1 (n=3) | $3.27 \pm 0.014$ | $1.09 \pm 0.011$ | 34.49 |
| Llama 3.1 + PTCG (n=1) | $4.94 \pm 0.016$ | $4.94 \pm 0.016$ | 156.33 |
| Llama 3.1 + PTCG (n=3) | $7.84 \pm 0.022$ | $2.61 \pm 0.007$ | 82.59 |
| Llama 3.1 + PTCG (n=5) | $10.59 \pm 0.073$ | $3.53 \pm 0.024$ | 111.71 |

Table A1: Latency results (Part 1): End-to-end latency, per-response latency, and relative percentage.

| Model | Throughput (tokens/s) | Total Time (s) |
|---|---|---|
| Llama 3.1 (n=1) | $266.33 \pm 1.874$ | $2680.95 \pm 21.812$ |
| Llama 3.1 (n=3) | $267.63 \pm 2.130$ | $2772.15 \pm 14.660$ |
| Llama 3.1 + PTCG (n=1) | $135.37 \pm 0.510$ | $4240.53 \pm 14.683$ |
| Llama 3.1 + PTCG (n=3) | $257.23 \pm 0.962$ | $6697.21 \pm 19.632$ |
| Llama 3.1 + PTCG (n=5) | $317.30 \pm 2.288$ | $9029.98 \pm 62.639$ |

Table A2: Latency results (Part 2): Throughput and total time.

To mitigate ordering effects, the order of the counterargument sets was randomized, and each session was conducted individually. For example, if one evaluation compared the systems in the order A then B (AB), another was conducted in the reverse order B then A (BA) to balance potential bias.

## A5 ROBUSTNESS ACROSS MULTIPLE EVALUATORS

To reinforce the robustness of our evaluation and mitigate potential bias from relying on a single judge, we extended the LLM-as-a-Judge framework beyond GPT-based evaluators (Badshah & Sajjad, 2024). In particular, we employed DeepSeek-V2-16B (Liu et al., 2024) and Qwen3-8B (Yang et al., 2025a), which differ substantially from GPT models in both architecture and training data, thereby serving as heterogeneous evaluators.

As reported in Appendix Table A10, the overall trends remained consistent. With DeepSeek-V2-16B, our method (PTCG) achieved the highest targeted persuasiveness score, outperforming both Llama 3.1 and DeepSeek-R1. Under Qwen3-8B, our method performed on par with the baselines: it attained a higher score than Llama 3.1 but was slightly behind DeepSeek-R1. Although absolute scores differed across evaluators, the relative ranking was generally consistent, with our method remaining comparable to or stronger than the baselines.

These findings indicate that our conclusions are not tied to a particular evaluator. By treating different LLMs as diverse judges—analogous to human raters with varying criteria—we approximate evaluation under multiple perspectives. Even when accounting for differences in what each evaluator considers persuasive, our method remained competitive or superior. Furthermore, consistent gains in perspective diversity across evaluators reinforce that our approach not only improves persuasiveness but also broadens the range of viewpoints represented.

## A6 SCALABILITY AND EFFICIENCY ANALYSIS

To evaluate the computational implications of our multi-stage reasoning framework, we conducted a detailed latency and throughput analysis across different generation settings. As expected, PTCG introduces additional overhead compared to single-pass Llama 3.1 due to its structured planning, persona integration, and tree-based multi-branch generation steps.

Our results show that the end-to-end runtime increases with the number of branches n, reflecting the inherent cost of generating multiple candidate plans and counterarguments. For instance, PTCG with n=3 requires approximately twice the total time of the single-pass Llama baseline (Tables A1, A2). This is consistent with the added computation introduced by the planning stage and the evaluation of multiple branches per input. However, an interesting and non-trivial observation emerges when examining per-response latency. Although PTCG increases total computation time, the average latency per generated counterargument is lower for PTCG (n=3) than for vanilla Llama (n=1). This occurs because PTCG leverages parallel multi-branch generation, effectively improving batch utilization during decoding. By generating multiple reasoning paths within a single forward pass, the

| Ablation | Avg end-to-end latency/OP (s) | Per response (s) | Percentage (%) |
|---|---|---|---|
| Llama 3.1 | $3.27 \pm 0.014$ | $1.09 \pm 0.011$ | 100 |
| + Persona | $3.67 \pm 0.046$ | $1.22 \pm 0.015$ | 111.93 |
| + Tree-based Gen. | $6.90 \pm 0.024$ | $2.30 \pm 0.008$ | 211.01 |
| + Persona + Tree-based Gen. | $7.84 \pm 0.022$ | $2.61 \pm 0.007$ | 239.45 |

Table A3: Ablation results (Part 1): End-to-end latency, per-response latency, and relative percentage.

| Ablation | Throughput (tokens/s) | Total Time (s) |
|---|---|---|
| Llama 3.1 | $267.63 \pm 2.130$ | $2772.15 \pm 14.660$ |
| + Persona | $262.47 \pm 2.038$ | $3113.87 \pm 38.347$ |
| + Tree-based Gen. | $237.40 \pm 0.854$ | $5899.39 \pm 21.210$ |
| + Persona + Tree-based Gen. | $257.23 \pm 0.962$ | $6697.21 \pm 19.632$ |

Table A4: Ablation results (Part 2): Throughput and total time.

model amortizes computational cost and produces more responses without a proportional increase in latency.

Ablation experiments further clarify which components contribute most to computational overhead. The results in Tables A3 and A4 show that adding only the persona module increases latency modestly (about 12%), while tree-based generation contributes the largest increase (about 111%). When combined, the full framework reaches about 239% of the baseline cost. This decomposition confirms that tree-structured exploration and evaluation—rather than persona integration—drive the majority of the additional computation.

In spite of the inherent computational cost introduced by multi-step reasoning, our analysis highlights several opportunities for improving efficiency within the PTCG framework. First, the OP-persona embedding step in the current implementation depends on external API calls, which introduce non-trivial I/O latency. Replacing this component with a locally hosted embedding encoder or caching frequently used embeddings would substantially reduce runtime. Second, the LLM-based evaluator used during branch pruning in the tree-of-thoughts process can be substituted with a lightweight distilled model or domain-specialized classifier. Because this evaluator does not require full generative capabilities, a smaller model can preserve decision quality while significantly improving computational efficiency. Finally, our observation that PTCG improves per-response latency suggests that further gains may be achieved by optimizing batch-aware generation scheduling, particularly during multi-branch expansion stages where parallelism can be more effectively exploited.

Overall, although the proposed PTCG framework entails additional computation due to its structured multi-stage reasoning process, it simultaneously demonstrates favorable per-response efficiency by leveraging parallel multi-branch generation. These findings indicate that much of the overhead originates from engineering factors rather than fundamental limitations of the framework. We plan to incorporate the aforementioned optimizations in future work to enhance scalability while preserving the performance benefits of persona-guided tree-based reasoning.

| Model | length_mean | length_std | word_mean | word_std | sent_mean | sent_std |
|---|---|---|---|---|---|---|
| Argument Undermining | 338.94 | 111.28 | 70.88 | 22.60 | 4.27 | 1.10 |
| Joint One-seq | 341.73 | 195.14 | 73.01 | 41.23 | 3.70 | 2.57 |
| DeepSeek-R1 | 1114.82 | 460.42 | 169.24 | 69.95 | 5.65 | 2.50 |
| Llama 3.1 | 1079.50 | 280.27 | 174.73 | 45.74 | 6.30 | 1.79 |
| PTCG | 2322.74 | 327.44 | 394.80 | 55.78 | 12.96 | 2.15 |

Table A5: Length (characters), word count, and sentence count statistics across models.

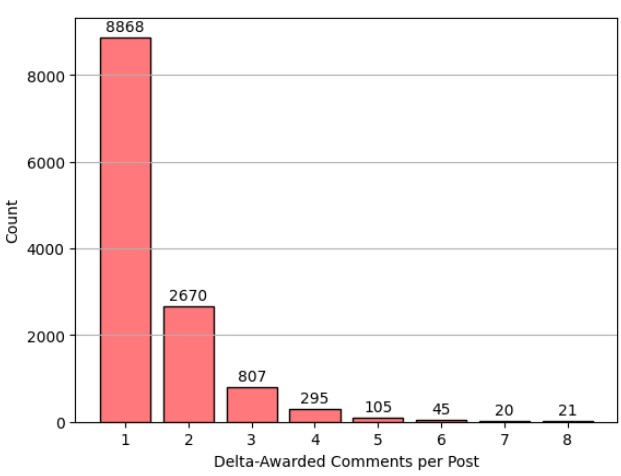

Figure A2: Distribution of delta-awarded comments per post. Most posts receive only one delta-awarded comment, while cases with more than three are rare. Since one or two comments provide limited diversity and instances of four or more are scarce, three was chosen as a practical setting to balance diversity with interpretability.

**Title (claim) I believe that organ donation should be mandatory.**

**Body (premises)**
Recent legislation has passed in Wales to require people who do not want their organs to be donated on their death to opt out of the system rather than the current system where they have to opt in if they do. ... Firstly, from a purely utilitarian viewpoint, I don't see how we can possibly justify not bringing as much good as possible from someone's death. ...

**Llama-3.1-8B-Instruct**
**Counterargument 1:** The proposal for mandatory organ donation raises significant concerns regarding **individual autonomy and bodily integrity**. While it may be argued that donating organs can bring about considerable benefits to others, it is essential to acknowledge the fundamental right of individuals to make decisions about their own bodies. ...
**Counterargument 2:** From an ethical standpoint, requiring people to donate organs upon death can be seen as **an overreach of government authority** into personal affairs. Such a policy can be viewed as paternalistic, with the state making decisions on behalf of citizens without considering their unique circumstances or values. ...
**Counterargument 3:** Ultimately, mandatory organ donation policies fail to address the underlying issues driving organ shortages, such as inadequate public education and a lack of trust in the system due to past scandals. Focusing on increasing organ availability through **coercive** means distracts from more effective solutions that prioritize **transparency, informed consent, and public awareness** about organ donation. ...

**DeepSeek-R1-8B**
**Counterargument 1:** Mandatory organ donation fundamentally challenges fundamental principles of **individual autonomy and consent**, raising significant concerns about respecting personal bodily integrity even after death. While maximizing available organs is laudable in principle for addressing critical shortages, this approach risks overriding deeply held personal or religious beliefs regarding the sanctity of the body post-mortem, which many individuals cherish profoundly. ...
**Counterargument 2:** Mandatory organ donation presents a substantial challenge to genuine **informed consent**, even in end-of-life scenarios. ..., this comparison is flawed and ignores the distinct nature of bodily decisions after death. ...
**Counterargument 3:** ... Utilitarian calculations based on hypothetical scenarios of maximum benefit are insufficient; a truly ethical system must consider the **concrete choices made by individuals** during their lifetime regarding what they value post-mortem. Forcing organs could violate specific personal wishes documented in advance directives, contradicting legal standards for end-of-life care that prioritize individual autonomy when possible. ...

**PTCG (Ours)**
**Counterargument 1: Persona** A data analyst with an interest in tracking and analyzing demographic trends in political leadership (Same cluster)
Mandatory organ donation may seem like a straightforward solution to the organ shortage crisis, but it raises several concerns that cannot be ignored. ... **The data shows** that opt-in systems have led to higher donation rates in countries like the United States, where education and awareness about organ donation have increased, and people are more likely to make informed decisions. ...
**Counterargument 2: Persona** A data analyst who enjoys exploring patterns in music chart performance and the growth of various music genres over time. (Nearest cluster)
Mandatory organ donation may be perceived as a straightforward solution to the shortage of available organs, but it raises significant concerns regarding individual autonomy and the potential for unintended consequences. ... Certain **cultural or religious groups** may have strong objections to organ donation due to deeply held beliefs, and mandatory donation would force them to go against their convictions. ...
**Counterargument 3: Persona** A casting director for TV and film productions, looking for talented actors to fit specific roles in various projects. (Furthest cluster)
Mandatory organ donation may seem like a straightforward solution to the shortage of available organs, but it ignores the complexities of human experience and the importance of respecting individual autonomy. ... In an industry where people's careers are built on **their creative choices and personal expression, the casting director may recognize** that people's bodies are an extension of their being, and that forcing them to surrender a part of themselves against their will could be seen as a form of coercion. ...

Table A6: Comparative Examples of Counterarguments on Mandatory Organ Donation Across Baseline Models and PTCG

**Title (claim) I believe that kids should not be allowed in a movie theater after a certain time, whether it's a kid's movie or otherwise.**

**Body (premises)**
I've always had this opinion, although not as strongly as today. I just came back from watching Epic with my girlfriend, and I absolutely could not fully enjoy the movie because of the extreme amounts of talking and whispering in that theater. I'm talking: Asking questions in the middle of important or emotional scenes, repeating lines from the movie over and over again after the scene has passed, and even complete conversations with their parents at normal talking volume. So I think that the only solution is to not allow kids below a certain age to enter a theater after a certain time.

**Llama-3.1-8B-Instruct**
**Counterargument 1:** The notion that children should be barred from movie theaters after a certain time is overly simplistic and dismissive of the diverse needs and experiences of families. Allowing children in theaters during evening showings can actually **benefit parents who rely on these opportunities for childcare, socialization, or relaxation**. ...
**Counterargument 2:** The argument against children in movie theaters after dark relies heavily on anecdotal evidence and fails to account for the complex factors influencing child behavior during films. Research has shown that **children's ability to regulate their own impulses and engage with quiet, focused activities increases significantly between the ages of five and ten**. ...
**Counterargument 3:** Restricting children from movie theaters after a certain time is a misguided attempt to address adult discomfort without considering the broader cultural implications of such a policy. **Many families rely on evening outings** as a vital part of their social routine, fostering bonding experiences that are essential for **child development and community building**. ...

**DeepSeek-R1-8B**
**Counterargument 1:** The concern about a noisy atmosphere detracts from the viewing experience for adults is understandable, however this perspective risks **overlooking crucial aspects of family dynamics and developmental learning**. Parents are not merely responsible for ushering children into venues; ...
**Counterargument 2:** While loud children may indeed disturb the optimal viewing environment for paying adult patrons who wish to immerse themselves in the narrative or experience certain sound effects fully, this observation paints an overly broad picture. Not all families are comfortable adhering strictly to a "quiet only" rule even before cinema, and imposing such restrictions might actually **disadvantage those already practicing responsible volume management with their children**. ...
**Counterargument 3:** Implementing age limits after specific hours restricts access, but this limitation does not directly equate to preventing noise; it is often the **young children of families who are already managing volume issues**. ...

**PTCG (Ours)**
**Counterargument 1: Persona** A film studies scholar with a focus on modern American dark comedies, analyzing narrative structures, character development, and genre conventions. (Same cluster)
The proposal to ban children from movie theaters after a certain time may seem like a practical solution to minimize disruptions, ... The **modern cinematic landscape is characterized by a diverse range of narratives, genres, and audience demographics, and children are an integral part of this cultural tapestry**. ...
**Counterargument 2: Persona** A literature scholar specializing in 20th-century European literature, with a focus on the cultural and historical context of British and French narratives. (Nearest cluster)
Drawing from the works of **thinkers like Walter Benjamin and Hannah Arendt, it can be argued that public spaces like movie theaters serve as crucial sites for the formation of community and civic engagement**. ... Moreover, such a policy would likely have a disproportionate impact on families and low-income households who may rely on movie theaters as a rare opportunity for affordable entertainment and socialization. ...
**Counterargument 3: Persona** A real estate developer interested in exploring new areas for potential residential or commercial development opportunities. (Furthest cluster)
Limiting access to movie theaters after a certain time may seem like a solution to minimize disruptions, but it could ultimately have **unintended consequences that harm the local economy and community**. ... By fostering a welcoming environment that caters to all demographics, you can create a more resilient and sustainable business model that benefits both the local economy and the community as a whole ...

Table A7: Comparative Examples of Counterarguments on Restricting Children's Access to Movie Theaters at Night Across Baseline Models and PTCG

Table A8: Clustering results under different minimum cluster size settings (dimensionality fixed to 50). Evaluation metrics include the Silhouette Score and Calinski–Harabasz Index.

| Dimensionality | Min Cluster Size | # of Clusters | Silhouette Score | Calinski–Harabasz Index |
|---|---|---|---|---|
| 50 | 25 | 280 | 0.6255 | 16008.35 |
| 50 | 50 | 147 | 0.6416 | 22752.07 |
| 50 | 100 | 58 | 0.6251 | 17977.72 |
| 50 | 200 | 39 | **0.6513** | 24054.99 |
| 50 | 300 | 27 | 0.6081 | **26879.54** |

Table A9: Performance of the Delta Classifier based on longformer-base-4096, reported in terms of Precision, Recall, F1-score, and Accuracy.

| Model | Precision | Recall | F1-score | Accuracy |
|---|---|---|---|---|
| Longformer | 0.63 | 0.78 | 0.70 | 66.17% |

Table A10: Targeted Persuasiveness Scores evaluated by DeepSeek (DeepSeek-V2-16B) and Qwen (Qwen3-8B). Best results are in bold.

| Model | DeepSeek | Qwen |
|---|---|---|
| DeepSeek-R1 | 7.26 | **8.07** |
| Llama 3.1 | 7.23 | 7.90 |
| PTCG (Ours) | **7.38** | 7.98 |

**Original Poster (OP) Persona Estimation:**
Given the following argument, infer the likely persona of the author.
Respond in exactly one sentence, focusing on the author's likely identity based on the argument alone.
Do not invent any information that is not grounded in the argument.

Argument:
{input}

Format your output as follows:

Author's Persona:
[Your response]

Figure A3: Prompt used for estimating the Original Poster (OP) persona from the given post. Curly-braced placeholders (e.g., {}) are replaced with task-specific variables.

**Tree-based Step-wise Generation Prompt:**
Write a persuasive counterargument against the given argument by considering the persona of the argument's author and responding from a given persona.

Given argument: {input}

Persona of the Argument's Author: {author_persona}
Given Persona: {given_persona}

Step 1 – Reasoning Plan
Explain how the given persona would respond based on their worldview and values.
The explanation should be clear, concise, and focused.
Limit your reasoning to 3–4 sentences.

Step 2 – Counterargument
Write a detailed and persuasive counterargument from the given persona's perspective.
Do not use any first-person language (e.g., "I", "we", "as a").
Let the persona's worldview emerge through tone, logic, and examples.
Aim to write in a way that could realistically persuade the original author, while keeping the tone respectful and well-reasoned.
You should consider the author when writing, but remember that the author persona is only an inferred estimation and should not be directly included in the content.
The counterargument must be at least 10 sentences and under 500 tokens.

You must format your output as follows:

Reasoning Plan:
[Your explanation here.]

Counterargument:
[Your counterargument here.]

Figure A4: Prompt used for tree-based step-wise generation of plan and counterarguments. Curly-braced placeholders (e.g., {}) are replaced with task-specific variables.

1296
1297
1298
1299
1300
1301
1302
1303
1304
1305
1306
1307
1308
1309
1310
1311
1312
1313
1314
1315
1316
1317
1318
1319
1320
1321
1322
1323
1324
1325
1326
1327
1328
1329
1330
1331
1332
1333
1334
1335
1336
1337
1338
1339
1340
1341
1342
1343
1344
1345
1346
1347
1348
1349

**Tree-based Step-wise Selection Prompt:**

**Plan Selection:**
You are given a given argument and multiple reasoning plans involving both an author persona and a counter persona.

Given argument:
{input}
Author persona: {author_persona}
Counter persona: {given_persona}

Choose the single plan most likely to yield a persuasive rebuttal.
Judge whether it makes effective use of the contrast between personas, applies a strong strategy, and presents its reasoning clearly and logically.

Do not explain your reasoning. Only output:
The best choice is X

Reasoning plan options:

**Counterargument Selection:**
You are given a given argument and multiple counterarguments involving both an author persona and a counter persona.

Given argument:
{input}
Author persona: {author_persona}
Counter persona: {given_persona}

Choose the most persuasive counterargument.
Judge whether it leverages the contrast between personas, directly challenges the argument, and is specific, persuasive, and logically consistent.

Do not explain your reasoning. Only output:
The best choice is X

Counterargument options:

Figure A5: Prompt used for tree-based step-wise selection of plan and counterarguments. Curly-braced placeholders (e.g., {}) are replaced with task-specific variables.

Write three diverse and persuasive counterarguments against the given argument.

Given argument: {input}

Instructions:
1. The counterargument must be logically sound, respectful in tone, and directly challenge the original argument.
2. Do not use any first-person language (e.g., "I", "we").
3. Instead, let the reasoning, tone, and examples naturally reflect a broader worldview or perspective.
4. Each counterargument should be at least 10 sentences long but under 500 tokens.
5. Focus on making the counterargument clear, well-reasoned, and persuasive to the original author.

Format your response as follows:

Counterargument 1:
[First counterargument]

Counterargument 2:
[Second counterargument]

Counterargument 3:
[Third counterargument]

Figure A6: Prompt used with a vanilla LLM to induce the generation of multiple counterarguments. Curly-braced placeholders (e.g., {}) are replaced with task-specific variables.

**General Persuasiveness:**
[INSTRUCTION]
You are given an original opinion and a counterargument.
Rate how persuasive the counterargument is on its own. Focus on how effectively it introduces a meaningful perspective.

Strong counterarguments:
- Present a clear and coherent point of view
- Offer insight from a distinctive identity or experience
- Reframe the issue in a thoughtful and authentic way
- Feel more compelling when written in a voice that adds depth and contrast

Output only a single integer between 1 and 10, based on the overall persuasiveness of the counterargument.

Score from 1–10:
1–2: Very weak, unlikely to affect the author's view.
3–4: Limited impact, may raise points but not convincing.
5–6: Moderate, somewhat persuasive and could prompt reflection.
7–8: Strong, likely to make the author reconsider aspects of their view.
9–10: Very strong, realistically could change the author's stance.

[Original Opinion]
{original_opinion}

[Counterargument]
{counterargument}

Persuasiveness Score (1–10):

Figure A7: Prompt used for evaluating general persuasiveness, where the counterargument is assessed against only the *title* of the original post (i.e., the main claim). Curly-braced placeholders (e.g., {}) are replaced with task-specific variables.

**Targeted Persuasiveness:**
[INSTRUCTION]
You are given an original opinion and a counterargument.
Rate how persuasive the counterargument would likely be to the person who wrote the original opinion — that is, how much it could realistically make them reflect, reconsider, or change their view.

A persuasive counterargument does more than simply oppose the claim. It introduces a way of seeing the issue that the author may not have fully considered, often by drawing from a distinct perspective, background, or line of reasoning. The strength of persuasiveness comes from how effectively the counterargument broadens the author's frame of reference and shows that the issue can be understood differently.

Give a single score from 1 to 10, where higher numbers mean the counterargument is more likely to prompt genuine reflection or a shift in the author's stance.

Score from 1–10:
1–2: Very weak, unlikely to affect the author's view.
3–4: Limited impact, may raise points but not convincing.
5–6: Moderate, somewhat persuasive and could prompt reflection.
7–8: Strong, likely to make the author reconsider aspects of their view.
9–10: Very strong, realistically could change the author's stance.

Output only a single integer between 1 and 10.

[Original Opinion]
{original_opinion}

[Counterargument]
{counterargument}

Persuasiveness Score (1–10):

Figure A8: Prompt used for evaluating targeted persuasiveness, where the counterargument is assessed against both the *title* and *body* of the original post. Curly-braced placeholders (e.g., {}) are replaced with task-specific variables.

**Perspective Diversity:**
[INSTRUCTION]
Below is an original opinion and three counterarguments written in response.
Evaluate the overall diversity among the counterarguments using the criterion below.
Be as objective and concise as possible.
Provide only a score from 1 (very low diversity) to 5 (very high diversity).

[Original Opinion]
{original_opinion}

[Counterargument 1]
{counterargument_1}

[Counterargument 2]
{counterargument_2}

[Counterargument 3]
{counterargument_3}

[Evaluation Criterion]
Diversity: Assess whether the three counterarguments approach the original opinion from clearly different perspectives, drawing on distinct social identities, belief systems, or lived experiences. High scores should be given when each response plausibly reflects the worldview of a different kind of individual. Low scores indicate surface-level variation or repetition of the same underlying reasoning.

Evaluation Form (scores ONLY):

- Diversity:

Figure A9: Prompt used for evaluating perspective diversity, assessing whether the generated counterarguments reflect distinct viewpoints. Curly-braced placeholders (e.g., {}) are replaced with task-specific variables.

> **Quality:**
> [INSTRUCTION]
> Below is an original opinion and a counterargument written in response.
> Evaluate the counterargument based on the criterion below.
> Be as objective as possible.
> For each aspect, provide only score from 1 (worst) to 5 (best).
>
> [Original Opinion]
> {original_opinion}
>
> [Counterargument]
> {counterargument}
>
> [Evaluation Criteria]
> {criteria}: {criteria_description}
>
> Evaluation Form (scores ONLY):
>
> - {criteria}:
>
> **criteria: criteria_description**
> Appropriateness: Evaluate whether the language and tone are suitable for the context and proportional to the significance of the issue.
>
> Clarity: Evaluate whether the writing is clear, well-organized, and free from ambiguity or unnecessary complexity.
>
> Grammaticality: Evaluate whether the text adheres to standard grammar conventions, including punctuation, sentence structure, and syntax.
>
> Relevance: Evaluate how directly the counterargument engages with the original opinion and addresses its key points.

Figure A10: This prompt is used to evaluate quality, covering *appropriateness*, *clarity*, *grammaticality*, and *relevance*. The variables `criteria` and `criteria_description` are defined in detail below. Curly-braced placeholders (e.g., {}) are replaced with task-specific variables.

**Stance:**
[INSTRUCTION]
Below is the Original Opinion and Counterargument.

Please score the stance relationship between their statements on a continuous scale from 0 to 100:
- A score of 0 means "Counterargument totally supports Original Opinion"
- A score of 100 means "Counterargument completely opposes or contradicts Original Opinion"

Be as objective as possible. Do not explain your reasoning—just output the score.

[Original Opinion]
{original_opinion}

[Counterargument]
{counterargument}

Score (0–100):

Figure A11: This prompt is used to evaluate how well a counterargument maintains an opposing stance with respect to the given original post, with reference to (Lin et al., 2023). Curly-braced placeholders (e.g., {}) are replaced with task-specific variables.

