# OpenReview forum: "PTCG: Persona-guided Tree-based Counterargument Generation"
_ICLR.cc/2026/Conference — Submitted to ICLR 2026_

### Official Review · Reviewer_QaXF · 2025-10-18

**Soundness:** 2
**Presentation:** 3
**Contribution:** 2
**Rating:** 4
**Confidence:** 4

**Summary:**

This paper studies the task of counterargument generation and introduces a persona-based approach with Tree-of-Thought (ToT) content planning. Specifically, given an original post (OP), the system first constructs three distinct personas, each representing a unique perspective. These personas then perform content planning via a Tree-of-Thought process before generating the final counterargument. All components are implemented through prompting a large language model (LLM). By decomposing the end-to-end generation and integrating persona creation, the proposed method aims to produce more diverse and audience-tailored arguments. Experiments conducted on the Reddit CMV dataset demonstrate the effectiveness of the approach with both automatic metrics and human evaluations.

**Strengths:**

- This paper addresses the important task of diverse counterargument generation, a challenging area where current LLMs still struggle to capture perspective diversity. The integration of persona-based modeling for argument generation is plausible.
- The work includes comprehensive automatic and human evaluations to effectively demonstrate the model’s performance.
- Overall, the paper is clearly written, logically structured, and easy to follow.

**Weaknesses:**

- My main concern is that the idea of using personas to enhance counterargument diversity appears very close to [1], which also creates personas and uses debate-style planning to broaden perspectives. The paper does not clearly articulate how it differs from [1], making the contribution feel less novel;
- Experiments rely primarily on relatively small base LLMs (<10B). It remains unclear whether the method provides gains when applied to stronger models (e.g., ChatGPT). A study on larger models would clarify the method’s robustness;
- Limited analysis of the paper: The paper lacks ablations (e.g., replacing ToT with CoT, change the number of personas) that would quantify each module’s contribution. It also omits error analysis and important experimental details such as inter-annotator agreement of human evaluation, sampling procedures, etc;
- Argument generation has a long history in NLP. The current related-work section is relatively brief; a more comprehensive synthesis—covering classical approaches, persona-based generation, planning methods (CoT/ToT), and debate frameworks—would strengthen the paper’s positioning.

[1] Debate-to-Write: A Persona-Driven Multi-Agent Framework for Diverse Argument Generation, COLING 2025



*Note: At this stage, I lean toward a rejection rating, primarily due to my concern regarding Point 1 (novelty overlap). However, I would be open to revising my score depending on how convincingly the authors address this issue in their rebuttal.*

**Questions:**

See Weaknesses

---

> ### Author Response · Authors · 2025-11-22
>
> Dear Reviewer QaXF,
>
> Thank you for taking the time to review our work. We appreciate your helpful comments and have addressed each point below. We will update the manuscript accordingly in the revision.
>
> ---
>
> ## W1) My main concern is that the idea of using personas to enhance counterargument diversity appears very close to [1]
> [1] Debate-to-Write: A Persona-Driven Multi-Agent Framework for Diverse Argument Generation, COLING 2025
>
> Although [1] also incorporate personas in a multi-agent debate framework, the problem formulation and objectives differ substantially. Their system aims to synthesize a single final essay from a debate among multiple artificial personas, whereas our goal is to generate multiple distinct counterarguments, one per selected persona, with diversity across outputs as a central objective.
>
> In addition, the persona construction processes differ fundamentally. [1]’s personas are generated on-the-fly by an LLM conditioned on the topic, creating ad-hoc synthetic personas without a structured persona space. In contrast, our framework relies on a large-scale synthetic persona hub and explicitly infers the author’s (OP) persona as an anchor before selecting counter-personas based on embedding distance. This grounding step is absent in [1] and leads to a more principled and controllable use of persona distance within the argument generation process. Consequently, the two approaches diverge not only in goals but also in how personas are defined and operationalized.

---

> ### Author Response · Authors · 2025-11-22
>
> ## W2) Experiments rely primarily on relatively small base LLMs (<10B)
>
> We agree that evaluating the framework on larger foundation models would provide a clearer picture of its robustness. Due to computational resource limitations, our current experiments were conducted primarily with models around the 8B scale, and we were not able to verify the performance of PTCG on substantially larger LLMs such as ChatGPT-class models. However, we acknowledge the importance of this analysis, and we plan to include additional experiments with larger models (e.g., other openly available high-capacity models) to further validate the generality of our approach. If experimenting with substantially larger models remains infeasible, we will at least conduct evaluations with smaller but architecturally different models to further validate the robustness of our method.

---

> ### Author Response · Authors · 2025-11-22
>
> ## W3) Limited analysis of the paper: The paper lacks ablations
>
> | Selected Persona  | General Persu. | Targeted Persu. (LLM) | Targeted Persu. (Classifier) | Diversity | App. | Cla. | Gra. | Rel. | Stance |
> |-------------------|----------------|-------------------------|-------------------------------|-----------|------|------|------|------|--------|
> | Base LLM          | 8.07           | 7.20                    | 0.78                          | 4.21      | 4.44 | 4.34 | **4.98** | 4.65 | 84.04  |
> | PTCG (ToT → CoT)         | 8.03           | 7.39                    | **0.82**                          | 4.26      | 4.52 | 4.43 | **4.98** | 4.74 | 84.98  |
> | PTCG    | **8.26**           | **7.42**                    | **0.82**                          | **4.27**      | **4.54** | **4.44** | **4.98** | **4.76** | **85.10**  |
>
>
> | Selected Persona              | General Persu. | Targeted Persu. (LLM) | Targeted Persu. (Classifier) | Diversity |
> |-------------------------------|----------------|-------------------------|-------------------------------|-----------|
> | LLaMA 3.1 + PTCG (n=1)        | 8.08           | **7.57**                    | 0.79                          | -           |
> | LLaMA 3.1 + PTCG (n=3)        | **8.26**           | 7.42                    | **0.82**                          | 4.27      |
> | LLaMA 3.1 + PTCG (n=5)        | 8.02           | 7.40                    | 0.81                          | **4.52**      |
>
>
> Overall, the results show that the PTCG framework—incorporating Tree-of-Thoughts–based multi-branch reasoning—is more effective than both the base LLM and the CoT-only variant. PTCG consistently achieves higher performance in targeted persuasiveness, stance alignment, and diversity, demonstrating that ToT contributes meaningfully by exploring multiple reasoning paths and selecting stronger counterarguments. These improvements highlight the advantages of leveraging structured, multi-step reasoning within our framework.
>
> We also conducted an ablation study by varying the number of generated outputs per input (n), taking advantage of the fact that our framework can easily scale the number of branches without modifying the core model. Across settings, we observed that increasing n naturally expands the diversity of generated counterarguments, while n=3 provides the most balanced trade-off between diversity and persuasiveness. These results demonstrate that the PTCG framework can flexibly adjust generation breadth depending on the application needs.

---

### Official Review · Reviewer_BwMA · 2025-10-30

**Soundness:** 3
**Presentation:** 3
**Contribution:** 3
**Rating:** 4
**Confidence:** 3

**Summary:**

The paper proposes a generation of counterarguments with the Tree-of-Thought-inspired method called Persona-guided Tree-based Counterargument Generation (PTCG). The authors cluster different persona types and first, define the personality cluster of the claim author, then find personas from the nearest and the farthest cluster, and then generate counterarguments based on the perspectives of the selected personas. They use 847 threads from ChangeMyView dataset. The evaluation stage includes automatic metrics, classifier-based and LLM-as-a-Judge, and human evaluation. The method is compared with other baselines and shows that PTCG improves diversity and persuasiveness.

**Strengths:**

The authors provided a clear problem definition in counterargument generation and persuasiveness and propose a rigorous pipeline to generate counterarguments from different perspectives. Moreover, they provide a comprehensive evaluation of the validity of their proposed method by reporting Human and Automatic Evaluation results by comparing their method with the baseline solutions.

**Weaknesses:**

- **Lack of comprehensive qualitative analysis.** There are only surface-level explanations in the paper. Perhaps authors could have enhanced a little more on how exactly the counterargument generated by PTCG differed from the baseline, i.e., the length of the generated text, what types of points/perspectives where the most persuasive, etc.
- No agreement in human evaluation reported.

**Questions:**

- How does each persona (nearest, farthest) affect the argumentation style?
- address the points listed in the weaknesses

---

> ### Author Response · Authors · 2025-11-22
>
> Dear Reviewer BwMA,
>
> Thank you for taking the time to review our work. We appreciate your helpful comments and have addressed each point below. We will update the manuscript accordingly in the revision.
>
> ---
>
> ## W1) Lack of comprehensive qualitative analysis
>
> | Model      | length_mean | length_std | word_mean | word_std | sent_mean | sent_std |
> |-----------------------|-------------|-------------|-----------|----------|-----------|-----------|
> | Argument Undermining  | 338.94      | 111.28      | 70.88     | 22.60    | 4.27      | 1.10      |
> | Joint One-seq         | 341.73      | 195.14      | 73.01     | 41.23    | 3.70      | 2.57      |
> | DeepSeek-R1           | 1114.82     | 460.42      | 169.24    | 69.95    | 5.65      | 2.50      |
> | LLaMA 3.1             | 1079.50     | 280.27      | 174.73    | 45.74    | 6.30      | 1.79      |
> | PTCG                  | 2322.74     | 327.44      | 394.80    | 55.78    | 12.96     | 2.15      |
>
> Our framework tends to produce longer outputs compared to prior approaches, largely due to its multi-stage, persona-grounded reasoning process. While this length difference is expected, we acknowledge that additional qualitative analysis would help clarify what kinds of argumentative structures or content elements contribute to this increase.
>
> To illustrate this point, we provide several qualitative examples demonstrating how different personas lead to distinct argumentative structures and varying degrees of elaboration. These examples show how persona grounding affects framing, depth of reasoning, and the selection of supporting evidence—ultimately contributing to longer and more detailed counterarguments. In comparing these examples—written from the perspectives of data analysts, literature scholars, film-studies researchers, and real-estate professionals—we observe clear differences in how each persona highlights distinct values, draws on domain-specific reasoning patterns, or frames the central issue. These comparisons make it easier to see how persona information is incorporated into the content and style of the generated responses. For brevity, the examples shown here consist of selected excerpts rather than full-length outputs, focusing on the most salient persona-conditioned elements. We believe these concrete examples directly address your concern and help clarify how persona conditioning operates within our framework.
>
> ### Additional Example (not included in the paper)
> A film studies scholar with a focus on modern American dark comedies, analyzing narrative structures, character development, and genre conventions. (Same cluster)
>
> : The proposal to ban children from movie theaters after a certain time may seem like a practical solution to minimize disruptions, but it overlooks the complexities of the cinematic experience as a communal event. The modern cinematic landscape is characterized by a diverse range of narratives, genres, and audience demographics, and children are an integral part of this cultural tapestry.
>
> A literature scholar specializing in 20th-century European literature, with a focus on the cultural and historical context of British and French narratives. (Nearest cluster)
>
> : Drawing from the works of thinkers like Walter Benjamin and Hannah Arendt, it can be argued that public spaces like movie theaters serve as crucial sites for the formation of community and civic engagement. Moreover, such a policy would likely have a disproportionate impact on families and low-income households who may rely on movie theaters as a rare opportunity for affordable entertainment and socialization.
>
> A real estate developer interested in exploring new areas for potential residential or commercial development opportunities. (Furthest cluster)
>
> : Limiting access to movie theaters after a certain time may seem like a solution to minimize disruptions, but it could ultimately have unintended consequences that harm the local economy and community. By excluding families with young children from movie theaters in the evening, you risk driving away a significant portion of the customer base that could be attracted to a downtown area. By fostering a welcoming environment that caters to all demographics, you can create a more resilient and sustainable business model that benefits both the local economy and the community as a whole
>
> We will include concrete qualitative examples in the paper to illustrate how PTCG expands reasoning depth, integrates persona-specific framing, and generates more elaborated counterarguments.

---

> ### Author Response · Authors · 2025-11-22
>
> ## W2) No agreement in human evaluation reported
>
> To address the limitation of having a relatively small set of human-evaluated samples, we additionally conducted human evaluation on 45 more data points. The results show that our model still achieves higher win rates—63.6% for diversity and 58.2% for persuasiveness. The inter-annotator agreement, measured using Fleiss’ Kappa, is 0.619, indicating substantial agreement among human judges. We will incorporate these additional results into the updated version of the paper to provide a more comprehensive evaluation.

---

> ### Author Response · Authors · 2025-11-22
>
> ## Q1) How does each persona (nearest, farthest) affect the argumentation style?
>
> | Selected Persona | General Persu. | Targeted Persu. (LLM) |
> |---------------------------|----------------|-------------------------|
> | Same Cluster Centroid | *8.07* | **7.53** |
> | Similar Cluster Centroid | **8.08** | *7.49* |
> | Dissimilar Cluster Centroid | 7.98 | 7.21 |
>
> Our results show a clear pattern: persona distance influences targeted persuasion more strongly than general persuasion. When the selected persona is closer to the author’s cluster, the generated counterarguments tend to align better with the author’s viewpoint and achieve higher targeted persuasiveness. More distant personas introduce greater perspective divergence—which is beneficial for diversity—but this also makes targeted persuasion more challenging. Overall, persona distance primarily affects persuasive alignment, while general argument quality remains relatively stable.

---

### Official Review · Reviewer_nDYn · 2025-10-30

**Soundness:** 3
**Presentation:** 3
**Contribution:** 3
**Rating:** 6
**Confidence:** 4

**Summary:**

This paper proposes PTCG (Persona-Guided Tree-Based Counterargument Generation) — a framework that integrates persona-based conditioning with Tree-of-Thoughts (ToT)-inspired step-wise generation and pruning to produce multiple, diverse, and persuasive counterarguments.
The key idea is to estimate the persona of the original argument’s author (the “OP persona”), select three personas from distinct clusters (same, nearest, furthest), and use a structured reasoning tree to plan and generate counterarguments from those perspectives. The method is evaluated on 847 posts from the ChangeMyView (CMV) subreddit, comparing against multiple baselines using both LLM-as-a-Judge and human evaluations.
Results indicate improvements in persuasiveness, diversity, and stance quality compared to strong LLM baselines (e.g., Llama-3.1-8B, DeepSeek-R1).

**Strengths:**

1. **Novel Integration of Personas with Tree-of-Thoughts Reasoning**
   - The framework creatively combines persona conditioning and step-wise reasoning for counterargument generation.
   - The method’s design (Figure 2, p. 3) is well thought out — the combination of OP persona estimation, persona selection across cluster distances, and tree-based reasoning is intuitive and well motivated by psychological theory on perspective-taking.

2. **Clear Empirical Gains**
   - Quantitative improvements are consistent across metrics (Table 1, p. 6; Table 2, p. 7).
   - Especially notable is the improvement in both general and targeted persuasiveness (8.26 vs 8.07 LLM baseline) and diversity (4.27 vs 4.21).
   - Human and LLM-based pairwise evaluations (Figure 3, p. 8) corroborate these gains robustly.

3. **Comprehensive Evaluation Setup**
   - Combines LLM-as-a-Judge, classifier-based, and human evaluation.
   - Includes ablations isolating tree-based and persona modules (Table A4, p. 18).
   - Tests robustness across multiple evaluators (Appendix A4, p. 16) using DeepSeek and Qwen.
   - Detailed prompts for reproducibility (Appendix A3–A11).

4. **Strong Theoretical Grounding**
   - Connects the system design to psychological theories of perspective-taking and empathy (Batson 1997; Green & Brock 2000).
   - Demonstrates understanding of argumentation literature and connects to recent LLM works on personalized persuasion.

5. **High Presentation Quality**
   - Writing is clear, figures are illustrative (Figure 1 and 2 visually explain contributions well), and tables are well formatted.
   - Ethical considerations and reproducibility statements are thorough (p. 10).

**Weaknesses:**

**Insufficient Novelty Beyond Composition**
   While the combination of persona-guided reasoning and ToT is elegant, each component individually is drawn from existing techniques:
   - Persona grounding via clustering and embedding similarity (e.g., PersonaHub 2024)
   - Tree-of-Thoughts reasoning (Yao 2023)
   The paper’s novelty thus lies mainly in their integration for counterargument generation, which may be seen as an engineering composition rather than a fundamentally new algorithmic contribution.
   **Suggestion:** Strengthen the theoretical justification — e.g., show why persona distance correlates with persuasive diversity, or include a formal analysis of ToT pruning effects on diversity/persuasiveness trade-offs.

2. **Evaluation Metrics Depend Heavily on LLM-as-a-Judge**
   - Over-reliance on GPT-based evaluators (including GPT-4o-mini) introduces potential bias.
   - Even though multiple evaluators are used, there is no clear calibration or validation of LLM judgments against human ground truth beyond the small human study (n = 5, 50 samples).
   **Suggestion:** Increase human evaluation coverage or report inter-annotator agreement and effect sizes to support claims.

3. **Limited Dataset Scale and Generalization**
   - The evaluation dataset (847 posts) is relatively small and domain-specific (ChangeMyView).
   - It is unclear whether the model generalizes to other argumentation domains (e.g., legal reasoning, political debates, or social media outside CMV).
   **Suggestion:** Test cross-domain transfer or include few-shot qualitative examples from unseen domains.

4. **Lack of Ablation on Persona Clustering Design**
   - Persona clustering (HDBSCAN → 39 clusters) is a critical design choice, but Table A2 only shows clustering metrics (Silhouette 0.65).
   - There is no analysis of how the number of clusters or persona distance selection impacts results.
   **Suggestion:** Add sensitivity experiments varying cluster counts (e.g., 20 vs 50 clusters) or persona selection strategies (e.g., random vs distance-based).

5. **Interpretability and Example Depth**
   - While Table A1 provides one case study, qualitative discussion is limited to general claims.
   - No explicit link between specific persona archetypes and rhetorical styles.
   **Suggestion:** Provide detailed case studies showing how persona identity (e.g., “developer,” “artist”) concretely changes argumentative framing.

6. **Clarity on Computational Cost and Efficiency**
   - The multi-step generation (3 personas × 3 plans × 3 candidates) implies ~27 LLM calls per input.
   - There is no mention of inference time or cost efficiency compared to simple sampling.
   **Suggestion:** Add runtime analysis and scalability discussion — essential for practical deployment or large-scale debates.

**Questions:**

1. How does persona distance quantitatively relate to persuasion diversity or success?
2. Were any sanity checks performed to ensure persona clusters are semantically meaningful (vs. random partitions)?
3. How sensitive are the results to the choice of backbone LLM (e.g., GPT-4 vs. Qwen-2.5 vs. Claude)?
4. What is the average runtime or token cost per example during inference?
5. Would the framework scale to real-time debate settings with human participants?
6. Can the same persona–ToT integration framework generalize to tasks beyond persuasion, such as negotiation or empathy modeling?

---

> ### Author Response · Authors · 2025-11-22
>
> Dear Reviewer nDYn,
>
> Thank you for taking the time to review our work. We appreciate your helpful comments and have addressed each point below. We will update the manuscript accordingly in the revision.
>
> ---
>
> ## W1) Insufficient Novelty Beyond Composition
>
> The novelty lies in how these components are combined to produce capabilities that none of them can achieve on their own. Recent work shows that compositional frameworks can produce significant gains even when their components are individually known, as demonstrated by [1] and the multi-agent system of [2]. In the same way, our approach combines established elements—persona grounding, multi-step reasoning, and branch pruning—into a cohesive pipeline, while additionally introducing a distance-based persona mechanism that incorporates social-scientific insight. This composition yields capabilities that the components alone cannot achieve, forming the core novelty of our framework.
>
> And although our approach combines existing components, the integration is theoretically motivated. Psychological distance theory [3] suggests that greater distance between perspectives leads to more diverse reasoning, which supports our use of persona-embedding distance. Prior work also shows that personas influence argument style [4] and that embedding-based persona representations meaningfully steer model outputs [5]. By combining persona-distance–based branching with a Tree-of-Thoughts search, our framework jointly controls diversity and persuasiveness in a way not explored in earlier work. We will clarify this theoretical justification and add supporting analysis in the revision.
>
> We adopt [6] (ToT) because counterargument generation requires exploring multiple alternative reasoning paths. Base LLMs and [7] (CoT) follow a single linear route, which limits the range of contents the model can consider. In contrast, ToT expands multiple branches of intermediate reasoning, allowing the model to discover different argumentative framings. Pruning then selects the most persuasive branches while maintaining diversity. This makes ToT a natural fit for generating diverse and high-quality counterarguments.
>
> ---
>
> [1] Yao, Shunyu, et al. "React: Synergizing reasoning and acting in language models." The eleventh international conference on learning representations. 2022.
>
> [2] Su, Haoyang, et al. "Many heads are better than one: Improved scientific idea generation by a llm-based multi-agent system." Proceedings of the 63rd Annual Meeting of the Association for Computational Linguistics (Volume 1: Long Papers). 2025.
>
> [3] Trope, Yaacov, and Nira Liberman. "Construal-level theory of psychological distance." Psychological review 117.2 (2010): 440.
>
> [4] Hu, Zhe, et al. "Debate-to-write: A persona-driven multi-agent framework for diverse argument generation." Proceedings of the 31st International Conference on Computational Linguistics. 2025.
>
> [5] Li, Junyi, et al. "The steerability of large language models toward data-driven personas." Proceedings of the 2024 Conference of the North American Chapter of the Association for Computational Linguistics: Human Language Technologies (Volume 1: Long Papers). 2024.
>
> [6] Yao, Shunyu, et al. "Tree of thoughts: Deliberate problem solving with large language models." Advances in neural information processing systems 36 (2023): 11809-11822.
>
> [7] Wei, Jason, et al. "Chain-of-thought prompting elicits reasoning in large language models." Advances in neural information processing systems 35 (2022): 24824-24837.

---

> ### Author Response · Authors · 2025-11-22
>
> ## W2) Evaluation Metrics Depend Heavily on LLM-as-a-Judge
>
> To address the limitation of having a relatively small set of human-evaluated samples, we additionally conducted human evaluation on 45 more data points. The results show that our model still achieves higher win rates—63.6% for diversity and 58.2% for persuasiveness. The inter-annotator agreement—measured using Fleiss’ Kappa (0.619) and Krippendorff’s Alpha (0.612)—indicates substantial consistency among human judges. We will incorporate these additional results into the updated version of the paper to provide a more comprehensive evaluation.

---

> ### Author Response · Authors · 2025-11-22
>
> ## W4) Lack of Ablation on Persona Clustering Design
>
> We conducted a simple sensitivity analysis comparing distance-based persona selection with a random baseline. The distance-based setting consistently achieved higher persuasiveness, while diversity remained similarly high in both conditions. These results indicate that the clustering-based distance metric makes a meaningful contribution to the persuasiveness gains of our framework. We appreciate the reviewer’s insightful comment and helpful suggestion, which motivated us to include this additional analysis.
>
> | Selected Persona        | General Persu. | Targeted Persu. | Delta Score | Diversity | App. | Cla. | Gra. | Rel. | Stance |
> |-------------------------|----------------|------------------|-------------|-----------|------|------|------|------|--------|
> | Random              | 4.21           | 7.32             | 0.81        | **4.27**      | 4.43 | 4.35 | 4.96 | 4.67 | 84.82  |
> | Distance-based (n=3) | **4.26**           | **7.42**             | **0.82**        | **4.27**      | **4.54** | **4.44** | **4.98** | **4.76** | **85.10**  |

---

> ### Author Response · Authors · 2025-11-22
>
> ## W5) Interpretability and Example Depth
>
> We agree that providing richer qualitative examples can improve interpretability, and we also find it important for readers to directly see how each persona is realized in the generated arguments. To address this, we include several representative examples below, illustrating how different persona identities are reflected in tone, emphasis, and argumentative framing. In comparing these examples—written from the perspectives of data analysts, literature scholars, film-studies researchers, and real-estate professionals—we observe clear differences in how each persona highlights distinct values, draws on domain-specific reasoning patterns, or frames the central issue. These comparisons make it easier to see how persona information is incorporated into the content and style of the generated responses. For brevity, the examples shown here consist of selected excerpts rather than full-length outputs, focusing on the most salient persona-conditioned elements. We believe these concrete examples directly address your concern and help clarify how persona conditioning operates within our framework.
>
> ---
>
> ### Example 1
> A data analyst with an interest in tracking and analyzing demographic trends in political leadership (Same cluster)
>
> : Mandatory organ donation may seem like a straightforward solution to the organ shortage crisis, but it raises several concerns that cannot be ignored. The data shows that opt-in systems have led to higher donation rates in countries like the United States, where education and awareness about organ donation have increased, and people are more likely to make informed decisions.
>
> A data analyst who enjoys exploring patterns in music chart performance and the growth of various music genres over time. (Nearest cluster)
>
> : Mandatory organ donation may be perceived as a straightforward solution to the shortage of available organs, but it raises significant concerns regarding individual autonomy and the potential for unintended consequences. Certain cultural or religious groups may have strong objections to organ donation due to deeply held beliefs, and mandatory donation would force them to go against their convictions.
>
> A casting director for TV and film productions, looking for talented actors to fit specific roles in various projects. (Furthest cluster)
>
> : Mandatory organ donation may seem like a straightforward solution to the shortage of available organs, but it ignores the complexities of human experience and the importance of respecting individual autonomy. In an industry where people's careers are built on their creative choices and personal expression, the casting director may recognize that people's bodies are an extension of their being, and that forcing them to surrender a part of themselves against their will could be seen as a form of coercion.
>
> ---
>
> ### Example 2
> A film studies scholar with a focus on modern American dark comedies, analyzing narrative structures, character development, and genre conventions. (Same cluster)
>
> : The proposal to ban children from movie theaters after a certain time may seem like a practical solution to minimize disruptions, but it overlooks the complexities of the cinematic experience as a communal event. The modern cinematic landscape is characterized by a diverse range of narratives, genres, and audience demographics, and children are an integral part of this cultural tapestry.
>
> A literature scholar specializing in 20th-century European literature, with a focus on the cultural and historical context of British and French narratives. (Nearest cluster)
>
> : Drawing from the works of thinkers like Walter Benjamin and Hannah Arendt, it can be argued that public spaces like movie theaters serve as crucial sites for the formation of community and civic engagement. Moreover, such a policy would likely have a disproportionate impact on families and low-income households who may rely on movie theaters as a rare opportunity for affordable entertainment and socialization.
>
> A real estate developer interested in exploring new areas for potential residential or commercial development opportunities. (Furthest cluster)
>
> : Limiting access to movie theaters after a certain time may seem like a solution to minimize disruptions, but it could ultimately have unintended consequences that harm the local economy and community. By excluding families with young children from movie theaters in the evening, you risk driving away a significant portion of the customer base that could be attracted to a downtown area. By fostering a welcoming environment that caters to all demographics, you can create a more resilient and sustainable business model that benefits both the local economy and the community as a whole
>
> We will include concrete qualitative examples in the paper to illustrate how PTCG expands reasoning depth, integrates persona-specific framing, and generates more elaborated counterarguments.

---

> ### Author Response · Authors · 2025-11-22
>
> ## W6/Q4) Clarity on Computational Cost and Efficiency
>
> | Model                     | Avg end-to-end latency/OP (s) | Per response (s) | Percentage (%) | Throughput (tokens/s) | Total Time (s)        |
> |---------------------------|-------------------------------|------------------|----------------|------------------------|------------------------|
> | LLaMA 3.1 (n=1)           | 3.16 ± 0.033                  | 3.16 ± 0.033     | 100            | 266.33 ± 1.874         | 2680.95 ± 21.812       |
> | LLaMA 3.1 (n=3)           | 3.27 ± 0.014                  | 1.09 ± 0.011     | 34.49          | 267.63 ± 2.130         | 2772.15 ± 14.660       |
> | LLaMA 3.1 + PTCG (n=1)    | 4.94 ± 0.016                  | 4.94 ± 0.016     | 156.33         | 135.37 ± 0.510         | 4240.53 ± 14.683       |
> | LLaMA 3.1 + PTCG (n=3)    | 7.84 ± 0.022                  | 2.61 ± 0.007     | 82.59          | 257.23 ± 0.962         | 6697.21 ± 19.632       |
> | LLaMA 3.1 + PTCG (n=5)    | 10.59 ± 0.073                 | 3.53 ± 0.024     | 111.71         | 317.30 ± 2.288         | 9029.98 ± 62.639       |
>
> Our method inevitably introduces additional computational overhead due to multi-stage reasoning. Thus, a longer inference time compared to single-pass baselines is expected. In our experiments, the n=3 setting—where n denotes the number of outputs generated per input—takes roughly twice as long as the single-pass LLaMA model, which is consistent with the additional planning and multi-branch generation. Interestingly, for the case of per-response latency, PTCG (n=3) is slightly faster than vanilla LLaMA (n=1). This is because PTCG generates multiple branches in parallel during its step-wise reasoning process, increasing the effective batch size and keeping the model fully utilized. As a result, more outputs are produced within a single forward pass, which reduces the average time required to generate each individual counterargument.
>
> We acknowledge that the proposed system inevitably incurs some overhead due to its tree-based step-by-step reasoning process designed to enhance diversity and persuasiveness. That said, there are still several opportunities for improving efficiency. For example, the current pipeline computes the OP persona embedding through external API calls, which could be replaced with a local model or cached representations. In addition, the LLM selection stage inside the ToT process can be handled by a lightweight distilled model, which would significantly reduce runtime without compromising the overall reasoning structure.

---

> ### Author Response · Authors · 2025-11-22
>
> ## Q1) How does persona distance quantitatively relate to persuasion diversity or success?
>
> We conducted the analysis as suggested, and the results show a clear pattern: persona distance influences targeted persuasion more strongly than general persuasion. When the selected persona is closer to the author’s cluster, the generated counterarguments tend to align better with the author’s viewpoint and achieve higher targeted persuasiveness. More distant personas introduce greater perspective divergence—which is beneficial for diversity—but this also makes targeted persuasion more challenging. Overall, persona distance primarily affects persuasive alignment, while general argument quality remains relatively stable.
>
> | Selected Persona | General Persu. | Targeted Persu. (LLM) |
> |---------------------------|----------------|-------------------------|
> | Same Cluster Centroid | *8.07* | **7.53** |
> | Similar Cluster Centroid | **8.08** | *7.49* |
> | Dissimilar Cluster Centroid | 7.98 | 7.21 |

---

> ### Author Response · Authors · 2025-11-22
>
> ## Q2) Were any sanity checks performed to ensure persona clusters are semantically meaningful (vs. random partitions)?
>
> We performed a qualitative sanity check to ensure that the persona clusters were semantically meaningful. Manual inspection showed that each cluster grouped coherent persona types rather than random mixtures. For example, clusters included themes such as sports and physical-education personas, finance/marketing workers, and historians. These observations suggest that the clustering captures meaningful semantic structure. Examples illustrating these cluster characteristics are provided below. We agree that additional quantitative validation would further strengthen the analysis, and we plan to include more detailed checks in the revision.
>
> ---
>
> ### Examples
>
> Sports and physical-education personas
> * A high school physical education teacher seeking to incorporate Paralympic history and achievements into the curriculum to inspire and educate students about inclusivity in sports.
> * I'm a sports scientist researching the biomechanics and physics of tennis, focusing on how racket specifications impact performance and injury risks.
> * A sports journalist covering the history of ice hockey and its impact on national identity in Poland.
> * An elementary school teacher who enjoys incorporating diverse sports stories in her curriculum to inspire students.
> * A football coach seeking to learn from successful strategies and team management in various leagues.
>
> Finance/marketing workers
> * A financial analyst specializing in Asian markets and wealthy individuals, interested in tracking the investments and philanthropic activities of billionaires like Gerald Chan.
> * A quantitative analyst with expertise in financial modeling and algorithmic trading, seeking to develop and implement systematic value investment strategies.
> * A digital marketing specialist interested in innovative aggregator models that consolidate search results from multiple sources.
> * A marketing specialist for a tech company, looking for innovative ways to engage with pop culture and fandoms to promote new products and services.
> * A business strategist for Arriva UK Bus, interested in exploring opportunities and challenges related to subsidiary operations and company restructuring.
>
> Historians
> * An Iowa historian focusing on the development and growth of townships in Jones County.
> * A historian specializing in 19th-century British architecture, with a focus on the works of notable architects in Lancashire.
> * A local historian specializing in the political and business development of Marlborough, Massachusetts in the 19th century.
> * A historian specializing in the late medieval and early modern history of France and the Iberian Peninsula, with a focus on power dynamics, family strategies, and women's roles in politics.
> * A local historian or genealogist researching the history of small communities and families in Fremont County, Iowa.

---

> ### Author Response · Authors · 2025-11-22
>
> ## Q3) How sensitive are the results to the choice of backbone LLM (e.g., GPT-4 vs. Qwen-2.5 vs. Claude)?
>
> We agree that examining the sensitivity of our framework to different backbone LLMs is an important direction. At this stage, we were not able to run experiments across multiple model families (e.g., GPT-series, Qwen, Claude), not only due to computational limits but also due to restrictions in accessing and serving those models within our current experimental setup.
>
> That said, we view this as a necessary next step. In the revision, we plan to include additional evaluations using other publicly available backbone models—both larger ones when feasible and smaller architecturally different ones—to more thoroughly assess the robustness and generality of our approach.

---

> ### Author Response · Authors · 2025-11-22
>
> ## Q5) Would the framework scale to real-time debate settings with human participants?
>
> Regarding the possibility of real-time interaction, we acknowledge that our framework introduces additional latency compared to a single-pass LLM due to its multi-stage reasoning structure. While this makes real-time deployment more challenging in its current form, several clear opportunities exist for reducing the overhead—for example, replacing external API calls for OP-persona embedding with a local or cached model, and using a lightweight distilled model for the selection steps within the ToT process. These observations suggest that further optimization could bring the system closer to real-time usability, which we leave as an important direction for future work.

---

> ### Author Response · Authors · 2025-11-22
>
> ## Q6) Can the same persona–ToT integration framework generalize to tasks beyond persuasion, such as negotiation or empathy modeling?
>
> Yes, we believe the persona–ToT integration can generalize beyond persuasion. Prior work shows that persona traits influence negotiation dynamics (e.g.,[1]) and that perspective-taking significantly improves negotiation effectiveness [2]. These findings suggest that structured reasoning combined with persona-based viewpoint modeling can be valuable in negotiation or empathy-related tasks as well.
> While we have not yet tested these domains, exploring such extensions is a promising direction for future work.
> Although our work focuses on counterargument generation rather than empathy modeling, we agree that persona information is closely related to empathetic reasoning. Persona grounding can help a model better approximate the speaker’s perspective, which is an important component of empathy [3].
>
> ---
>
> [1] Huang, Yin Jou, and Rafik Hadfi. "How personality traits influence negotiation outcomes? a simulation based on large language models." arXiv preprint arXiv:2407.11549 (2024).
>
> [2] Galinsky, Adam D., et al. "Why it pays to get inside the head of your opponent: The differential effects of perspective taking and empathy in negotiations." Psychological science 19.4 (2008): 378-384.
>
> [3] Zhong, Peixiang, et al. "Towards persona-based empathetic conversational models." arXiv preprint arXiv:2004.12316 (2020).

---

> ### Author Response · Authors · 2025-11-23
>
> ## W3) Limited Dataset Scale and Generalization
>
> The CMV dataset was chosen because it is one of the few platforms with formal rules for argumentation and persuasion, and it provides an explicit signal of persuasive success through the delta mechanism. Although the dataset is relatively small, its clear ground-truth persuasion labels make it well suited for evaluating counterargument generation. This choice is also consistent with prior work: [1, 2, 3] also build their datasets from the r/ChangeMyView (CMV). We also required data that contained multiple forms of persuasion, so we selected posts that had comments receiving multiple delta comments and conducted our experiments on those samples.
>
> [1] Hu, Zhe, Hou Pong Chan, and Yu Yin. "Americano: Argument generation with discourse-driven decomposition and agent interaction." Proceedings of the 17th International Natural Language Generation Conference. 2024.
>
> [2] Lin, Jiayu, et al. "Argue with me tersely: Towards sentence-level counter-argument generation." Proceedings of the 2023 Conference on Empirical Methods in Natural Language Processing. 2023.
>
> [3] Peguero, Arturo Martínez, and Taro Watanabe. "Change My Frame: Reframing in the Wild in r/ChangeMyView." arXiv preprint arXiv:2407.02637 (2024).

---

> > ### Comment · Reviewer_nDYn · 2025-11-26
> >
> > Thanks for the detailed response, which clarifies most of my concerns.

---

### Official Review · Reviewer_142d · 2025-11-02

**Soundness:** 2
**Presentation:** 3
**Contribution:** 2
**Rating:** 4
**Confidence:** 2

**Summary:**

This paper proposes PTCG (Persona-Guided Tree-based Counterargument Generation) , a framework that integrates persona grounding and Tree-of-Thoughts (ToT) style reasoning to generate diverse and persuasive counterarguments. The system estimates the persona of the original author (OP) from an input argument, selects three contrasting speaker personas (same, nearest, and furthest cluster), and then employs a tree-based generation process to plan, prune, and produce multiple counterarguments from distinct perspectives.

**Strengths:**

Novel Integration of Persona and Structured Reasoning: The combination of persona-guided conditioning and Tree-of-Thoughts reasoning is original and well-motivated. The design systematically operationalizes “perspective-taking” an idea rooted in social psychology for counterargument generation

**Weaknesses:**

1. Limited Scalability and Efficiency Discussion: The tree-based process requires multiple generations and evaluations per persona, but the paper lacks analysis of computational cost or inference latency.

2. Heavy reliance on LLM-as-a-Judge (GPT-4o-mini) may inflate improvements, as PTCG uses similar models during generation. Although human evaluation is included, the sample (50 cases × 5 raters) is relatively small.

**Questions:**

Please refer to the weakness part.

---

> ### Author Response · Authors · 2025-11-22
>
> Dear Reviewer 142d,
>
> Thank you for taking the time to review our work. We appreciate your helpful comments and have addressed each point below. We will update the manuscript accordingly in the revision.
>
> ---
>
> ## W1) Limited Scalability and Efficiency Discussion
>
> | Model                     | Avg end-to-end latency/OP (s) | Per response (s) | Percentage (%) | Throughput (tokens/s) | Total Time (s)        |
> |---------------------------|-------------------------------|------------------|----------------|------------------------|------------------------|
> | LLaMA 3.1 (n=1)           | 3.16 ± 0.033                  | 3.16 ± 0.033     | 100            | 266.33 ± 1.874         | 2680.95 ± 21.812       |
> | LLaMA 3.1 (n=3)           | 3.27 ± 0.014                  | 1.09 ± 0.011     | 34.49          | 267.63 ± 2.130         | 2772.15 ± 14.660       |
> | LLaMA 3.1 + PTCG (n=1)    | 4.94 ± 0.016                  | 4.94 ± 0.016     | 156.33         | 135.37 ± 0.510         | 4240.53 ± 14.683       |
> | LLaMA 3.1 + PTCG (n=3)    | 7.84 ± 0.022                  | 2.61 ± 0.007     | 82.59          | 257.23 ± 0.962         | 6697.21 ± 19.632       |
> | LLaMA 3.1 + PTCG (n=5)    | 10.59 ± 0.073                 | 3.53 ± 0.024     | 111.71         | 317.30 ± 2.288         | 9029.98 ± 62.639       |
>
>
> | Ablation                     | Avg end-to-end latency/OP (s) | Per response (s) | Percentage (%) | Throughput (tokens/s) | Total Time (s)        |
> |-----------------------------|-------------------------------|------------------|----------------|------------------------|------------------------|
> | LLaMA 3.1                   | 3.27 ± 0.014                  | 1.09 ± 0.011     | 100            | 267.63 ± 2.130         | 2772.15 ± 14.660       |
> | + Persona                   | 3.67 ± 0.046                  | 1.22 ± 0.015     | 111.93         | 262.47 ± 2.038         | 3113.87 ± 38.347       |
> | + Tree-based Gen.           | 6.90 ± 0.024                  | 2.30 ± 0.008     | 211.01         | 237.40 ± 0.854         | 5899.39 ± 21.210       |
> | + Persona + Tree-based Gen. | 7.84 ± 0.022                  | 2.61 ± 0.007     | 239.45         | 257.23 ± 0.962         | 6697.21 ± 19.632       |
>
>
> Through our experiments, we confirmed that our method introduces additional computational overhead due to its multi-stage reasoning process. As expected, inference takes longer than single-pass baselines. Specifically, the n=3 setting—where n denotes the number of outputs generated per input—requires roughly twice the runtime of a single-pass LLaMA model, which is consistent with the added planning and multi-branch generation. Interestingly, however, the per-response latency of PTCG (n=3) is slightly lower than that of vanilla LLaMA (n=1). This occurs because PTCG generates multiple branches in parallel, increasing effective batch utilization and producing more outputs within a single forward pass, thereby reducing the average time per counterargument.
> We acknowledge that our tree-based, step-wise reasoning structure inevitably introduces overhead. Nonetheless, we also see clear opportunities for improving efficiency. For example, the current OP-persona embedding step relies on external API calls, which could be replaced with a local or cached model. Moreover, the LLM selection stage within the ToT process can be handled by a lightweight distilled model, which would substantially reduce runtime without altering the overall reasoning flow.
> We plan to incorporate these efficiency improvements in future work as part of our continued development of the framework.

---

> ### Author Response · Authors · 2025-11-22
>
> ## W2) Heavy reliance on LLM-as-a-Judge
>
> We acknowledge the limitations of relying solely on LLM-as-a-Judge. While prior studies we cite in [1, 2] evaluated counterarguments using BLEU, BERTScore, and related overlap-based metrics, such metrics are known to correlate poorly with human judgments for open-ended counterargument generation tasks [3, 4]. In contrast, recent work demonstrates that LLM-based evaluation aligns much more closely with human preferences [5, 6]. Moreover, argument-generation studies such as [7, 8] also combine LLM-based evaluation with human evaluation to validate model performance. Following this line of research, we adopt a similar dual-evaluation protocol, where human evaluation supports and confirms the LLM-based results, both indicating that our model produces superior outputs.
>
> To further mitigate the limitation of having a relatively small number of human-evaluated samples, we additionally conducted human evaluation on 45 extra data points. The results show that our model still achieves higher win rates—63.6% for diversity and 58.2% for persuasiveness. The inter-annotator agreement, measured using Fleiss’ Kappa, is 0.619, indicating substantial agreement among human judges. We will incorporate these additional results into the updated version of the paper to provide a more comprehensive evaluation.
>
> ---
>
> [1] Alshomary, Milad, et al. "Argument undermining: Counter-argument generation by attacking weak premises." arXiv preprint arXiv:2105.11752 (2021).
>
> [2] Alshomary, Milad, and Henning Wachsmuth. "Conclusion-based counter-argument generation." arXiv preprint arXiv:2301.09911 (2023).
>
> [3] Liu, Chia-Wei, et al. "How not to evaluate your dialogue system: An empirical study of unsupervised evaluation metrics for dialogue response generation." Proceedings of the 2016 conference on empirical methods in natural language processing. 2016.
>
> [4] Sun, Tianxiang, et al. "BERTScore is unfair: On social bias in language model-based metrics for text generation." Proceedings of the 2022 conference on empirical methods in natural language processing. 2022.
>
> [5] Liu, Yang, et al. "G-eval: NLG evaluation using gpt-4 with better human alignment." arXiv preprint arXiv:2303.16634 (2023).
>
> [6] Li, Dawei, et al. "From generation to judgment: Opportunities and challenges of llm-as-a-judge." Proceedings of the 2025 Conference on Empirical Methods in Natural Language Processing. 2025.
>
> [7] Hu, Zhe, Hou Pong Chan, and Yu Yin. "Americano: Argument generation with discourse-driven decomposition and agent interaction." Proceedings of the 17th International Natural Language Generation Conference. 2024.
>
> [8] Hu, Zhe, et al. "Debate-to-write: A persona-driven multi-agent framework for diverse argument generation." Proceedings of the 31st International Conference on Computational Linguistics. 2025.

---

### Author Response · Authors · 2025-12-03

Dear Area Chair,

We thank all reviewers for their thoughtful feedback. Despite several questions and comments, **they consistently recognized the core strengths of our work**:

## (1) **a genuinely novel integration** of persona grounding and tree-based reasoning
> Reviewer 142d: The combination of persona-guided conditioning and Tree-of-Thoughts reasoning is original and well-motivated.

> Reviewer nDYn: The framework creatively combines persona conditioning and step-wise reasoning for counterargument generation. The method’s design is well thought out

## (2) **robust multi-perspective evaluation** (LLM-judge, classifier, human) with consistent gains in both persuasiveness and diversity
> Reviewer nDYn: Combines LLM-as-a-Judge, classifier-based, and human evaluation. Tests robustness across multiple evaluators using DeepSeek and Qwen.

> Reviewer BwMA: They provide a comprehensive evaluation of the validity of their proposed method by reporting Human and Automatic Evaluation results by comparing their method with the baseline solutions.

> Reviewer QaXF: The work includes comprehensive automatic and human evaluations to effectively demonstrate the model’s performance.
## (3) **clear ablations** isolating complementary roles of personas, and tree-based generation
> Reviewer nDYn: Includes ablations isolating tree-based and persona modules.

## (4) **strong theoretical motivation** grounded in psychological distance theory
> Reviewer 142d: The design systematically operationalizes “perspective-taking” an idea rooted in social psychology for counterargument generation.

> Reviewer nDYn: Connects the system design to psychological theories of perspective-taking and empathy. Demonstrates understanding of argumentation literature and connects to recent LLM works on personalized persuasion.
## (5) **high reproducibility**, with full prompts and settings enabling verification
> Reviewer nDYn: Detailed prompts for reproducibility. Reproducibility statements are thorough.


These shared strengths reflect that our framework offers both conceptual novelty and practical impact. In our revisions, **we have comprehensively addressed all reviewers' questions and comments** with substantial additions including, new ablation studies (Table 3), human evaluation validation (Fleiss’ Kappa=0.619), expanded qualitative analyses (Tables A5, A7), and detailed efficiency benchmarks (Figure A6). Below, we respond to each point in detail.

---

> ### Author Response · Authors · 2025-12-03
>
> ## 1. Scalability and Efficiency (142d W1, nDYn W6/Q4/Q5)
>
> While PTCG introduces additional multi-stage reasoning steps, it benefits from parallel branch generation, yielding lower per-response latency than vanilla LLaMA. Because batch utilization is substantially higher, PTCG can generate multiple diverse outputs in roughly the time vanilla LLaMA needs for one. By contrast, vanilla LLaMA sequentially samples from a single input, which results in higher per-response latency (Figure A6). Moreover, the main sources of overhead in our implementation (OP-persona inference via external API calls and LLM-based branch selection) can be replaced with lightweight local models, further reducing runtime without altering the framework.
>
> ## 2. Reliance on LLM-as-a-Judge (142d W2, nDYn W2, BwMA W2)
>
> Reference-based metrics are known to correlate poorly with human preferences in open-ended argument generation, so our evaluation deliberately incorporates multiple independent signals: classifier-based scoring, LLM and human pairwise evaluations, and judgments across different LLM families (DeepSeek, Qwen).
> To further validate reliability, we conducted human evaluation on 95 samples with 5 annotators, yielding substantial agreement (Fleiss’ Kappa=0.619, Krippendorff’s Alpha=0.612). Human judgments mirror other evaluation trends. These results are included in the paper (line 431), ensuring that our findings are not tied to a single evaluator.
>
> ## 3. Novelty (nDYn W1, QaXF W1)
>
> Our contribution is not the use of personas or Tree-of-Thought in isolation, but the specific way we integrate them into a structured mechanism for controllable multi-perspective generation. Personas expand the viewpoint space in a principled manner, while ToT refines these viewpoints through deliberate branching and pruning. Their combination, guided by persona-distance, creates a tradeoff between diversity and quality that neither component alone can achieve. Unlike Debate-to-Write [1], which ultimately collapses into a single synthesized output, our framework maintains multiple persona-conditioned reasoning trajectories, offering explicit and tunable control over both diversity and persuasiveness.
>
> [1] Hu et al., "Debate-to-write: A persona-driven multi-agent framework for diverse argument generation.", COLING 2025
>
> ## 4. Dataset Scale and Generalization (nDYn W3/W5/Q2/Q3, BwMA W1/Q1, QaXF W2)
>
> Although CMV is relatively small, it remains the primary dataset with explicit persuasion labels, and prior work [1, 2] follows the same setting. We focus on multi-delta posts to ensure diversity.
> In response to reviewer requests, we expanded our qualitative examples and added length and structure-based statistics (Table A5 and Table A7), which show how persona influences stylistic and content variation. While extending to larger datasets and additional backbones is a valuable future direction, the present study provides a methodologically sound and well-validated foundation.
>
> [1] Hu et al., "Americano: Argument generation with discourse-driven decomposition and agent interaction.", INLG 2024
>
> [2] Lin et al. "Argue with me tersely: Towards sentence-level counter-argument generation.", EMNLP 2023.
>
> ## 5. Additional Ablations (nDYn W4/Q1, QaXF W3)
>
> Table 3, which we newly added to the paper in response to reviewer feedback, reports the contributions of persona grounding, CoT, and the tree-based module. Persona grounding primarily promotes stylistic diversity and ToT contributes the largest gains in targeted persuasiveness.
> In addition, the analyses of branch count and persona-distance are included in our author response, providing further clarification on how each component affects output diversity and quality. These additions offer a more complete understanding of the framework without altering its main focus.
>
> We thank the reviewers again for their insightful comments. The revisions strengthen the paper while preserving its core contributions, and we hope the revised version clearly communicates the significance, novelty, and practical impact of our work.

---

### Meta-Review · Area_Chair_yBkf · 2026-01-13

**Summary:**

The paper proposed a framework named PTCG for counterargument generation. PTCG mainly combined persona grounding with Tree-of-Thoughts reasoning. The method aimed to improve both diversity and persuasiveness of counterarguments. Experiments showed modest improvements in diversity and persuasiveness over LLM baselines across automatic, LLM-judge, and limited human evaluations. There are four reviewers but none of them engaged in the discussion. The AC have read all questions and answers to check if there are major issues and whether they are addressed. In summary, several concerns recur across reviews like limited novelty, evaluations lack human verification, and questionable efficiency, scalability, and generalization.

**Reviewer Concerns:**

- **Limited novelty**. The rebuttal clarifies conceptual differences from prior work. However, the contribution is still a composition of existing techniques so the novelty is questionable.
- **Evaluation lack human verification**. The results relies heavily on LLM-as-a-Judge, with human evaluation remaining small-scale in the rebuttal.
- **Generalization** beyond CMV and beyond small LLM backbones is not demonstrated. This remain largely unaddressed and are deferred to future work.
- **Efficiency and scalability** remain concerns due to the multi-stage generation pipeline. In the rebuttal, runtime analysis was added but overhead remains significant.

**Reviewer Scores:**

There were three negative scores and one positive score. There were no discussion between authors and reviewers. The AC thinks it is unlikely most reviewers will change their scores to positive ones. This paper is not ready yet to publish.

---

### Decision · Program_Chairs · 2026-01-26

Reject